# A New Fractional-Order Load Frequency Control for Multi-Renewable Energy Interconnected Plants Using Skill Optimization Algorithm

Ahmed Fathy [1,2,*], Hegazy Rezk [3], Seydali Ferahtia [4], Rania M. Ghoniem [5], Reem Alkanhel [5] and Mohamed M. Ghoniem [6]

1 Electrical Engineering Department, Faculty of Engineering, Jouf University, Sakaka 72388, Saudi Arabia
2 Electrical Engineering Department, Faculty of Engineering, Zagazig University, Zagazig 44519, Egypt
3 Department of Electrical Engineering, College of Engineering in Wadi Alddawasir,
  Prince Sattam Bin Abdulaziz University, Al-Kharj 11942, Saudi Arabia
4 Laboratoire de Genie Electrique, Department of Electrical Engineering, University of M'Sila,
  M'Sila 28000, Algeria
5 Department of Information Technology, College of Computer and Information Sciences,
  Princess Nourah bint Abdulrahman University, Riyadh 11671, Saudi Arabia
6 Department of Computer, Mansoura University, Mansoura 35516, Egypt
* Correspondence: afali@ju.edu.sa

**Abstract:** Connection between electric power networks is essential to cover any deficit in the generation of power from any of them. The exchange powers of the plants during load disturbance should not be violated beyond their specified values. This can be achieved by installing load frequency control (LFC); therefore, this paper proposes a new metaheuristic-based approach using a skill optimization algorithm (SOA) to design a fractional-order proportional integral derivative (FOPID)-LFC approach with multi-interconnected systems. The target is minimizing the integral time absolute error (ITAE) of frequency and exchange power violations. Two power systems are investigated. The first one has two connected plants of photovoltaic (PV) and thermal units. The second system contains four plants, namely, PV, wind turbine, and two thermal plants, with governor dead-band (GDB) and generation rate constraints (GRC). Different load disturbances are analyzed in both considered systems. Extensive comparisons to the use of chef-based optimization algorithm (CBOA), jumping spider optimization algorithm (JSOA), Bonobo optimization (BO), Tasmanian devil optimization (TDO), and Atomic orbital search (AOS) are conducted. Moreover, statistical tests of Friedman ANOVA table, Wilcoxon rank test, Friedman rank test, and Kruskal Wallis test are implemented. Regarding the two interconnected areas, the proposed SOA achieved the minimum fitness value of 1.8779 pu during 10% disturbance on thermal plant. In addition, it outperformed all other approaches in the case of 1% disturbance on the first area as it achieved ITAE of 0.0327 pu. The obtained results proved the competence and reliability of the proposed SOA in designing an efficient FOPID-LFC in multi-interconnected power systems with multiple sources.

**Keywords:** LFC; PV plant; wind energy; multi-interconnected system; renewable energy; skill optimization algorithm

## 1. Introduction

Achieving stability of power system operation is essential to guarantee the continuity of customer services, especially during load disturbances. The frequency of the power system is greatly affected by the load violation. This problem arises in multi-interconnected systems, as the load violation causes a change in both frequency and exchange of power between the connected plants. Therefore, load frequency control (LFC) is essential to banish the violations in both frequency and tie-line power [1–3]. Many researchers have dealt with designing the LFC in multi-area multi-sources, which may be conventional or renewable

energy. Yousri et al. [4] introduced Harris Hawks optimization (HHO)-based methodology to obtain the optimal gains of LFC-proportional-integral (PI) inserted in interconnected systems with renewable energy sources (RESs). The authors considered the integral time absolute error (ITAE) as the target to be minimized. Ali et al. [5] recommended multiverse optimizer (MVO) as an efficient tool to design the model predictive control (MPC) LFC inserted in six-interconnected systems with renewables-based plants with storage systems. Moreover, the effect of generation rate constraint (GRC) and governor dead band (GDB) zones for thermal plant have been considered. Fathy et al. [6] designed the fractional-order proportional integral derivative (FOPID) LFC via movable damped wave algorithm (MDVA) to banish the variations of frequency deviations and tie-line powers of an interconnected system with RESs during load disturbance. Moreover, ITAE was the target in the considered optimization problem. In [7], the optimal gains of PI and PID-LFC have been determined via particle swarm optimizer (PSO); the designed controller has been used with hybrid solar-wind-micro-hydro interconnected systems. An extensive review of many approaches employed in designing LFC with solar-wind interconnected systems has been conducted in [8]. Additionally, the authors designed the FOPID-controller via the flower pollination algorithm (FPA). Fathy et al. [9] presented LFC simulated via an adaptive neuro fuzzy inference system (ANFIS) trained by antlion optimizer (ALO) installed in two and four-interconnected power plants with renewable energy sources. The authors considered the ITAE of frequency and tie-line power deviations as the fitness function to be minimized. Many reported techniques assigned to design LFC insertion in classical and modern power systems have been reviewed in [10], including nonlinear models, controller parameter identification, soft computing approaches, integration of renewable energy sources, future trends, and challenges. In [11], LFC designed via decentralized MPC has been installed within the Egyptian power system with traditional and renewable-based plants. The considered renewable-based plants were the wind farms of Zafarana and Gabel El-Zeit, and Benban solar plant. Additionally, both GRC and GDB of thermal plants have been considered. In [12], the authors identified the optimal parameters of proportional-derivative LFC with a cascaded filter via coyote optimization algorithm (COA); the controller has been used with renewable energy-based plants. Arora et al. [13] introduced an approach-based on moth flame optimizer (MFO) to minimize the frequency constraints of renewable energy generation systems via installing LFC. A comprehensive review of different LFC structures in both single and multi-interconnected systems has been reported in [14–17]. Takayama et al. [18] introduced a coordination method between the LFC and economic dispatching control (EDC) for large-scale renewable energy generation plants. Additionally, the optimal size of a battery storage system was analyzed based on the presented LFC and EDC. An improved twin delayed deep deterministic policy gradient deep reinforcement learning-based LFC has been introduced and incorporated in RESs with variable loads and electric vehicles [19]. The integral absolute error (IAE) of the frequency and exchange power deviations was selected as the target to be minimized. In [20], the marine predators algorithm (MPA) has been presented to find the parameters of LFC-PID inserted in interconnected systems with RESs and storage systems. An enhanced COA has been employed to design PI-PI and PD with filter-based LFC installed with RESs-based interconnected system [21]. The Ziegler–Nichols method has been used by Subham et al. [22] to adjust the parameters of automatic LFC-PID inserted in hybrid power systems. Moreover, hardware-in-loop based OP4510 has been simulated to assess the presented controller. PI, PID, and fuzzy-based LFC have been presented to mitigate the frequency and tie-line power violations of interconnected systems with renewable energy sources [23]. In [24], a hybrid approach comprising anopheles search algorithm and artificial intelligence techniques has been introduced to design PID-LFC incorporated in a triple interconnected hybrid system including solar, biomass, and fuel cell-based plants. Masuta et al. [25] presented a coordinated LFC for conventional power plants, battery energy storage system, heat pump water heater, and electric vehicles in addition to renewable energy sources. Dutta et al. [26] introduced an emotional controller for LFC inserted in a two area hybrid

power system with solar and biomass generating sources in addition to electric vehicles. Salp swarm algorithm (SSA) has been used to tune the parameters of LFC-PID inserted in a multi-area hybrid power system with renewable energy sources [27]. A sliding mode-based LFC has been incorporated in a multi-area interconnected system with integrated RESs [28]; moreover, the authors investigated the system asymptotic stability via Lyapunov theory on the basis of a linear matrix inequality technique. In [29], the authors identified the optimal gains of FOPID-LFC using an improved chaotic atom search optimizer (IASO); the controller was inserted in a multi-area system with multi-hybrid sources such that the ITAE of the frequency and exchange power violations is minimized. A multi-area hybrid source interconnected system including photovoltaics (PV), diesel engines, micro hydro generating units, and fuel cells as storage system has been established and controlled via fuzzy logic-based LFC [30]. Three types of LFC controllers were introduced in [31]: cascaded fractional order controller, three degrees of freedom PID, and tilt integral derivative one. The presented controllers have been installed in an interconnected power system with wind generation systems. Moreover, equilibrium optimizer (EO) has been employed to tune the gains of different controllers. A fractional order two degrees of freedom-based LFC has been designed using Quasi-oppositional HHO to achieve stable frequency for two identical area power systems including PV, biogas unit, wind turbine, and thermal power plant [32]. Xu et al. [33] presented an artificial sheep algorithm-based LFC installed in an interconnected-area with RESs. In [34], the author used teaching learning-based optimizer (TLBO) to get the optimal parameters of automatic LFC incorporated in a multi-source system of thermal, hydro, and gas plants. The authors used different fitness functions such as integral squared error (ISE), IAE, integral time squared error (ITSE), and ITAE to assess the presented approach. A reference offset governor approach has been used by Tedesco et al. [35] to simulate the LFC in a multi-area microgrid with renewable energy sources. A power control approach of hybrid renewable energy systems has been introduced in [36]. A parallel buck-boost converter controlled via fuzzy logic control has been constructed and installed in a hybrid renewable energy-based system [37].

The reported methods employed many metaheuristic optimizers and other artificial intelligence techniques, such as fuzzy logic and ANFIS, in designing the LFC. Many of the used optimizers are unable to obtain the desired results due to getting stuck in local optima. Moreover, the others need several parameters that should be defined by the user. Furthermore, the artificial intelligence-based approaches are not accurate due to the imprecise defined data, and also, they require excessive data for training. All these shortcomings are taken in consideration when conducting the present analysis.

The aim of this work is to design a FOIPD controller based LFC via a recent approach using the skill optimization algorithm (SOA). The algorithm is responsible for identifying the unknown parameters of the considered controller such that the integral time absolute error of the frequency and exchange power deviations is minimized.

The work contributions can be listed as follows:

- A new skill optimization algorithm (SOA)-based methodology is proposed to design FOPID-LFC installed with interconnected systems with RESs.
- Two power systems are investigated, PV/thermal and thermal/wind turbine/thermal/PV, at different load disturbances.
- An extensive comparison of CBOA, JSOA, BO, TDO, and AOS is conducted.
- Statistical tests of Friedman ANOVA table, Wilcoxon rank test, Friedman rank test, and Kruskal Wallis test are implemented.
- The competence and reliability of the proposed SOA are confirmed via the obtained results.

The paper is organized as follows: Section 2 presents the mathematical model of the interconnected system. Section 3 explains the main principle of a fractional-order PID controller (FOPID). The proposed skill optimization algorithm is presented in Section 4, while Section 5 introduces a formulation using the proposed optimization problem. The numerical analysis is given in Section 6, while Section 7 handles the conclusions.

## 2. Model of Interconnected Systems

Two multi-interconnected multi-sources systems are constructed and analyzed in this work. The first one covers two areas with photovoltaic (PV) with maximum power point tracker (PV) and thermal plants. The second system has four-interconnected plants comprising PV, wind turbine (WT), and two thermal units with GDB and GRC. Many reported works have been conducted to model PV and WT-based generating systems [38–40]. The considered systems are investigated under different load disturbances, and the mathematical model of each plant is presented in this section.

### 2.1. PV Plant Model

The PV generation system comprises solar cells that may be configured in series and/or parallel to generate the required power and cover the customer's needs. The generated power from the PV system is affected by the variation in weather conditions such as irradiance and temperature. Hence, the PV panel output voltage can be written as follows [41]:

$$V_{pv} = \left(\frac{n_s \varepsilon k T}{Q}\right) ln \left(\frac{n_p G I_{ph} - I + n_p I_o}{n_p I_o}\right) - \left(\frac{n_s I R_s}{n_p}\right) \tag{1}$$

where $I$ and $V_{pv}$ denote the PV panel output current and voltage, $n_s$ and $n_p$ are the number of series and parallel cells, $\varepsilon$ is the factor of completion, $k$ represents the Boltzmann constant, $T$ is the PV panel temperature, $Q$ represents the electron charge, $G$ is the irradiance in W/m$^2$, $I_{ph}$ and $I_o$ are the photo and saturation currents, and $R_s$ denotes the cell series resistance. The characteristics of the PV panel are nonlinear. The PV panel power-voltage (P-V) curve has a unique global maximum power (GMP) as shown in Figure 1, and it is essential to monitor this point to enhance the PV panel performance and maximize its efficiency. This target is achieved via MPPT which tunes the duty cycle of the DC-DC converter at the PV panel terminals. Many approaches have been used in simulating the MPPT; one of the most popular is called incremental conductance (INC) [42], which is used in this work.

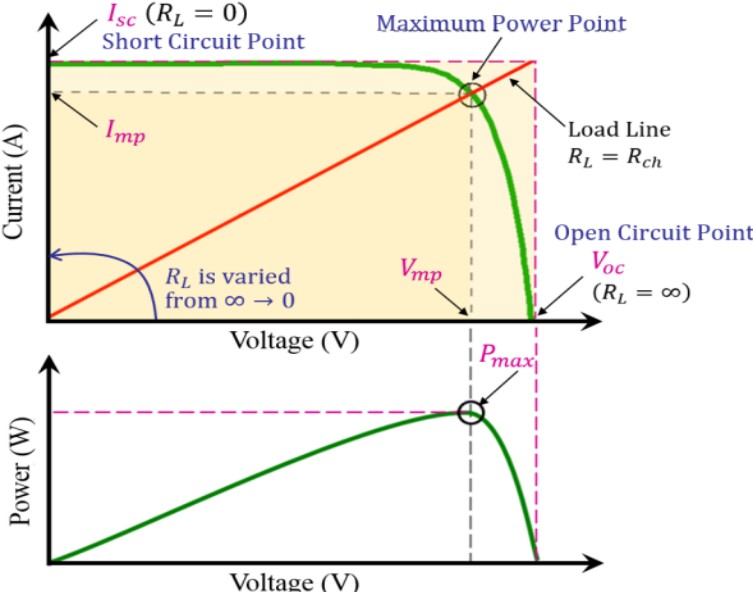

**Figure 1.** I-V and P-V characteristics of PV panel.

This approach depends on the derivative of the PV power to the voltage in three considered regions which are GMP right, GMP, and GMP left as follows:

$$\begin{cases} \frac{dP_{pv}}{dV_{pv}} > 0 & At\ right\ of\ GMP \\ \frac{dP_{pv}}{dV_{pv}} = 0 & At\ GMP \\ \frac{dP_{pv}}{dV_{pv}} > 0 & At\ left\ of\ GMP \end{cases} \tag{2}$$

where $P_{pv}$ represents the PV module output power. Finally, the PV generating unit including PV panel, DC-DC converter, MPPT, and inverter can be represented as two cascaded blocks with two gains as follows [43]:

$$G_1 = \left( \frac{S^2}{S^2 + \omega^2} \right) \left( \frac{V_{pv}(S^2 + \omega^2)(S^2 + 2\omega^2)}{kS^2(S^2 + 4\omega^2)} \right) \left( \frac{1 - e^{-ST_s}}{ST_s} \right) \tag{3}$$

$$G_2 = \left( \frac{\frac{M_1}{LC}}{S^2 + \left(\frac{1}{RC}\right)S + \frac{1}{LC}} \right) \left( \frac{1 - e^{\frac{-ST_s}{2}}}{1 + e^{\frac{-ST_s}{2}}} \right) \left( \frac{M_2}{1 + ST_s} \right) \tag{4}$$

where $\omega$ represents the angular frequency of the grid, $R$, $C$, and $L$ denote the output resistance, capacitance, and inductance of the converter, $T_s$ represents the simulation time, $M_1$ and $M_2$ represent the voltage gains of the buck converter and inverter, respectively.

*2.2. Model of WT*

The wind turbine (WT) operation is characterized by the coefficient of power ($C_p$); this depends on important parameters known as the tip ratio ($\lambda$) and the pitch angle of blade ($\beta$). The WT output power can be maximized by keeping the value of the tip speed ratio at its optimum value. The value of $\lambda$ can be calculated as [44],

$$\lambda = \frac{\omega_t r}{V_w} \tag{5}$$

where $\omega_t$ is the mechanical angular speed of the turbine, $r$ denotes the radius of the turbine, and $V_w$ is the wind speed. The wind power can be calculated as [45],

$$P_W = \frac{1}{2}\rho A C_p(\lambda, \beta) V_W^3 \tag{6}$$

where $\rho$ is the air density and $A$ denotes the swept area of turbine blades. The value of the power coefficient can be expressed as follows:

$$C_p = (0.44 - 0.0167\beta) \sin\left( \frac{\pi(\lambda - 2)}{15 - 0.3\beta} \right) - 0.00184(\lambda - 3)\beta \tag{7}$$

The wind turbine output power can be calculated as [46],

$$P_m = C_p(\lambda, \beta) P_W \tag{8}$$

The WT output power variations versus the rotor speed are shown in Figure 2. The wind plant transfer function can be expressed as [47],

$$G_{WT}(s) = \left( \frac{K_{pw1}(1 + sT_{pw1})}{1 + s} \right) \left( \frac{K_{pw2}}{1 + sT_{pw2}} \right) \left( \frac{K_{pw3}}{1 + s} \right) \tag{9}$$

where $K_{pw1}$, $K_{pw2}$, and $K_{pw3}$ denote the wind plant gains while $T_{pw1}$, $T_{pw2}$, and $T_{pw3}$ are the wind plant time constants.

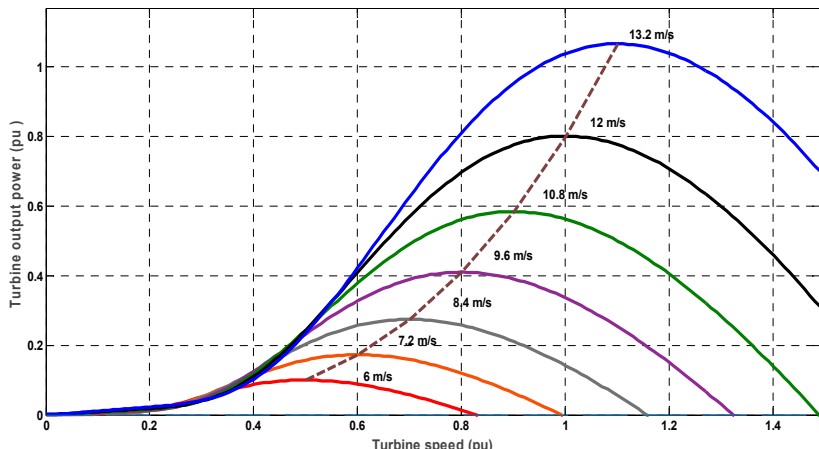

**Figure 2.** WT output power versus the rotor speed.

### 2.3. Thermal Plant Model

A thermal generating unit contains a steam turbine as prime mover, speed governor, reheater, and generating unit. The transfer functions of all these components can be written as follows [48]:

$$G_t = \frac{K_t}{1 + T_t S} \tag{10}$$

$$G_g = \frac{K_g}{1 + T_g S} \tag{11}$$

$$G_r = \frac{1 + K_r T_r S}{1 + T_r S} \tag{12}$$

$$G_{gen} = \frac{K_p}{1 + T_p S} \tag{13}$$

where $G_t$, $G_g$, $G_r$, and $G_{gen}$ denote the turbine, governor, reheater, and generator transfer functions, respectively, $K_t$, $K_g$, $K_r$, and $K_p$ denote the gains of turbine, governor, reheater, and generator, respectively, $T_t$, $T_g$, $T_r$, and $T_p$ are the time constants of the stated components, respectively. As stated before, the authors considered two interconnected multi-source power systems. The first one is a PV/thermal system, and Figure 3 shows the block diagram of the connected system. The second system is thermal/WT/thermal/PV system; both GDB and GRC of thermal units are considered; the architecture of such a system is shown in Figure 4.

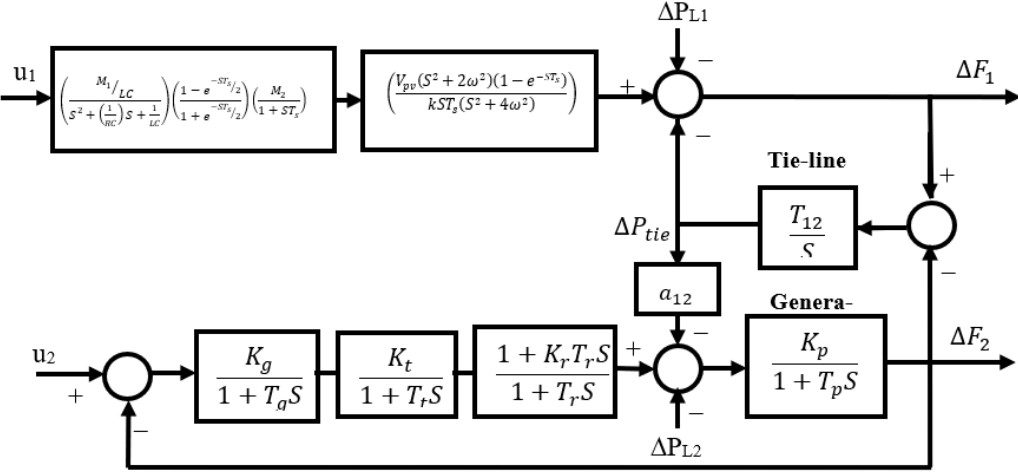

**Figure 3.** The considered PV/thermal connected power system.

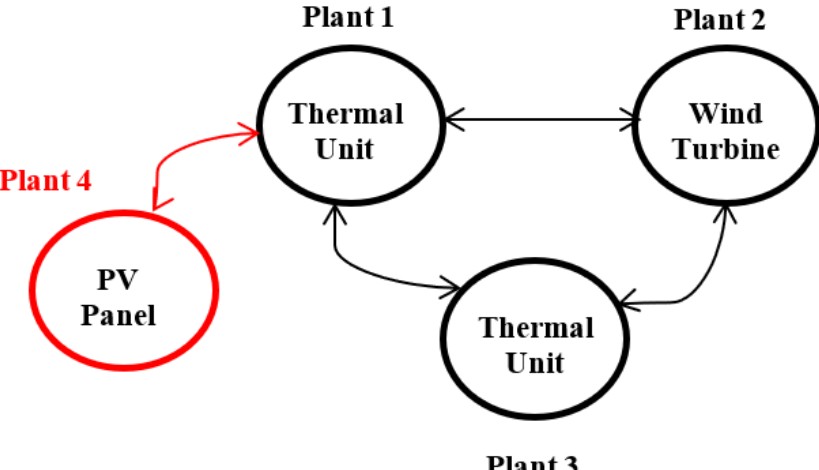

**Figure 4.** The architecture of thermal/WT/thermal/PV system.

### 3. Fractional-Order PID Controller (FOPID)

A fractional-order PID controller (FOPID) was presented in [49]; it has superior performance compared to the conventional PID for closed-loop systems. FOPID is different from the conventional PID as the order of its integral and derivative is not an integer. This gives the controller more freedom in the controller tuning, resulting in a better dynamic performance of FOPID than the conventional PID. Five parameters are used to identify such a controller, $k_p$, $k_i$, $k_d$, $\lambda_d$, and $\mu$, and the controller transfer function can be expressed as,

$$G_c = k_p + \frac{k_i}{S^{\lambda_d}} + k_d S^{\mu}, \lambda, \mu > 0 \tag{14}$$

By solving Equation (12), one can get the controller output as follows:

$$u(t) = k_p e(t) + k_i D^{-\lambda_d} e(t) + k_d D^{-\mu} e(t) \tag{15}$$

where $e(t)$ denotes the input of FOPID; the controller is inserted before the plant to feed it with the reference input such that the error between the reference input and the plant actual output is minimized. The FOPID performance can be enhanced via tuning the parameters $\lambda_d$ and $\mu$. which is This is the approach proposed here.

The FOPID controller is simulated in Simulink via FOMCON Toolbox. It depends on fractional-order calculus to model, design, and control the system. There are block sets provided by this toolbox from which $PI^{\lambda} D^{\mu}$ (FOPID) controller can operate.

### 4. The Proposed Skill Optimization Algorithm

The skill optimization algorithm (SOA) was introduced by Givi1 et al. [50]. It is inspired by human efforts to develop enhanced skills. In life, the people (members of SAO) strive to improve their skills via learning. The approach begins by initializing the members randomly. The population matrix of SOA can be formulated as follows:

$$X = \begin{bmatrix} X_1 \\ \vdots \\ X_i \\ \vdots \\ X_N \end{bmatrix} = \begin{bmatrix} x_{1,1} & \cdots & x_{1,d} & \cdots & x_{1,m} \\ \vdots & \vdots & \vdots & \vdots & \vdots \\ x_{i,1} & \cdots & x_{i,d} & \cdots & x_{i,m} \\ \vdots & \vdots & \vdots & \vdots & \vdots \\ x_{N,1} & \cdots & x_{N,d} & \cdots & x_{N,m} \end{bmatrix} \tag{16}$$

where $X_i$ is the $i$th candidate, $x_{i,d}$ denotes the $d$th variable value proposed via the $i$th member of the population, $N$ represents the number of members, and $m$ denotes the number of

considered variables. Each row in the population matrix represents the candidate solution; the fitness function of each one is calculated and expressed as follows:

$$F = \begin{bmatrix} F_1 \\ \vdots \\ F_i \\ \vdots \\ F_N \end{bmatrix} = \begin{bmatrix} F(X_1) \\ \vdots \\ F(X_i) \\ \vdots \\ F(X_N) \end{bmatrix} \qquad (17)$$

where $F_i$ denotes the fitness value of the $i$th candidate solution; the best fitness value recognizes the best member and vice versa. During the iterative process followed in SOA, the fitness values are updated in addition to the worst and best members. Two phases are followed in SOA to update the population members, which are exploration and exploitation. In the first stage, the skill learning process is conducted via experts, while the second stage depends on the activities and individual efforts. The main objective of the exploration phase is to search for the global solution in the search space via moving the algorithm members under the guidance of other members. The original optimal area is identified properly when the exploration power is increased. On the other hand, the exploitation phase aims at local search, which helps in converging better solutions.

*4.1. Exploration Phase*

In this phase, the population member strives to enhance his skill via an expert member who has a good condition based on fitness value. The expert member is selected randomly from the members with better fitness values than the considered member. It guides the population member to different locations in the search space via learning the skill to do this. The new location is accepted when its fitness value is improved, this can be modeled as follows:

$$x_{i,d}^{P1} = x_{i,d} + rand \times (E_{i,d} - I \times x_{i,d}), E_i = X_k if \ F_k < F_i \qquad (18)$$

$$x_{i,d} = \begin{cases} x_{i,d}^{P1} & if \ F_i^{P1} < F_i \\ x_{i,d} & if \ F_i^{P1} \geq F_i \end{cases} \qquad (19)$$

where $x_{i,d}^{P1}$ denotes the updated position of member $i, d$ in the first phase (P1), $F_i^{P1}$ is the fitness value of $x_{i,d}^{P1}$, $E_i$ represents the selected expert member, $E_{i,d}$ is the dth dimension of the expert member, *rand* is a random number in range [0, 1], and $I$ represents a random number that has a value of either 1 or 2.

*4.2. Exploitation Phase*

In this phase, each member strives to enhance the skill gained in the exploration phase via individual activity and practice. It is modeled as a local search to increase the exploitation such that the member tries to enhance his fitness value. This can be conducted as,

$$x_{i,d}^{P2} = \begin{cases} x_{i,d} + \frac{1 - 2 \times rand \times x_{i,d}}{t} & if \ rand < 0.5 \\ x_{i,d} + \frac{LB_j + rand(UB_j - LB_j)}{t} & if \ rand < 0.5 \end{cases} \qquad (20)$$

$$x_{i,d} = \begin{cases} x_{i,d}^{P2} & if \ F_i^{P2} < F_i \\ x_{i,d} & if \ F_i^{P2} \geq F_i \end{cases} \qquad (21)$$

where $x_{i,d}^{P2}$ is the updated position of member $i, d$ in the second phase (P1), $F_i^{P2}$ represents the fitness value of $x_{i,d}^{P2}$, $t$ is the number of iterations, $UB_j$ and $LB_j$ are the upper and lower limits of the $j$th variable. The steps followed in SOA are given in Figure 5.

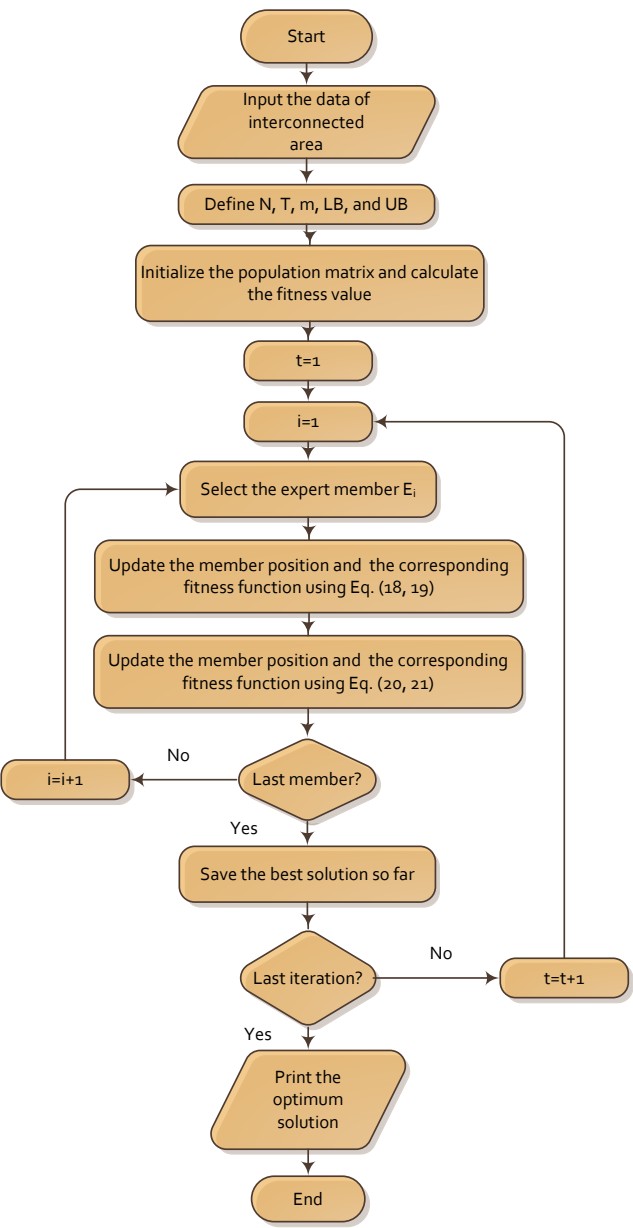

**Figure 5.** SOA flowchart.

## 5. The Proposed Optimization Problem

The problem of designing an LFC inserted in interconnected RESs is formulated as an optimization problem with a constrained fitness function, the formulation of the proposed optimization problem and the solution methodology is presented in this section.

### 5.1. The Fitness Function

In this work, the authors considered the integral time absolute error (ITAE) of frequency and exchange power violations as the target to be minimized. ITAE integrates the time multiplied by absolute error over a specified time; the ITAE tuning crops systems that settle abundantly more rapidly than the other tuning methods that use IAE and ISE. The considered variables to be identified are the FOPID parameters, $k_p$, $k_i$, $k_d$, $\lambda_d$, and $\mu$.

The fitness function can be written as follows [6]:

$$Minimize\ ITAE = \int_0^t \left( \left( \sum_{i=1}^{n_a} |\Delta F_i + \Delta P_{tie,i}| \right).t \right) dt \tag{22}$$

where $\Delta F_i$ and $P_{tie,i}$ are the violations in *i*th area frequency and exchange power, respectively, *t* denotes the specified time, and $n_a$ is the number of interconnected plants. The constraints accompanying the optimization problem can be written as,

$$
\begin{aligned}
k_p{}^{min} &\leq k_p < k_p{}^{max} \\
k_i{}^{min} &\leq k_i < k_i{}^{max} \\
k_d{}^{min} &\leq k_d < k_d{}^{max} \\
\lambda_d{}^{min} &\leq \lambda_d < \lambda_d{}^{max} \\
\mu^{min} &\leq \mu < \mu^{max}
\end{aligned}
\tag{23}
$$

where *min* and *max* denote the minimum and maximum limits of the scaling variable, and have been set in the range 0.1–2 [51].

### 5.2. The SOA-Based Solution Methodology

The SOA-based methodology is responsible for identifying the optimal parameters of FOPID, resulting in minimum ITAE. The controller with the best parameters has a small input signal ($e(t)$) which is the difference between the plant′s actual output ($y(t)$) and reference input ($w(t)$). The controller feeds the power plant with the required signal that helps in mitigating the violations in both frequency and exchange power. The configuration of a power plant with the proposed SOA-FOPID is shown in Figure 6. The adapted parameters of the FOPID controller are $k_p$, $k_i$, $k_d$, $\lambda_d$, and $\mu$. They are identified via the proposed SOA to minimize the error signal ($e(t)$).

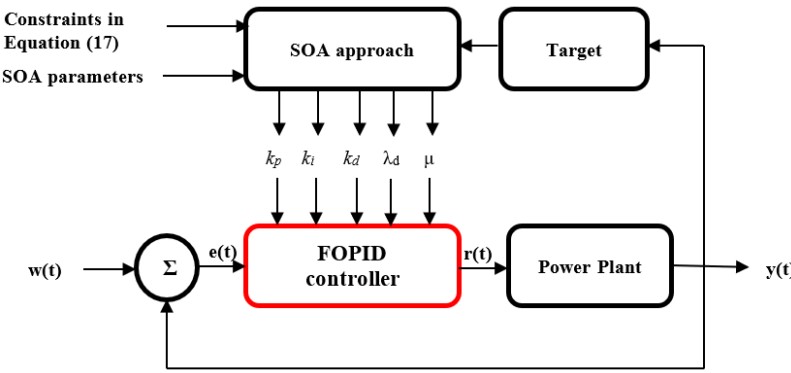

**Figure 6.** The configuration of a power plant with the proposed SOA-FOPID.

## 6. Numerical Analysis

In this work, the designed FOPID via SOA is investigated via two interconnected multi-source power systems. The first system is a PV plant connected to a thermal generating unit. The second system contains four power systems, namely, two thermal generation plants, a wind turbine, and PV. Additionally, the effect of GDB and GRC are considered in the thermal plants. Furthermore, different load disturbances have been analyzed in both considered systems. The proposed SOA is compared to other approaches using the chef-based optimization algorithm (CBOA), jumping spider optimization algorithm (JSOA), Bonobo optimization (BO), Tasmanian devil optimization (TDO), and Atomic orbital search (AOS). All approaches are implemented for 50 population size, 100 iterations, and 10 independent runs.

### 6.1. Two-Interconnected Power System

The two-part interconnected power system with the proposed FOPID is modeled in Simulink/Matlab. The system configuration is shown in Figure 7. The system parameters given in [9] are used in this work. The PV plant and thermal generator capacities are 500 MW and 2000 MW, respectively. The adapted parameters of the FOPID controller installed with the PV plant are $k_{p1}$, $k_{i1}$, $k_{d1}$, $\lambda_{d1}$, and $\mu_1$, while those of the controller of the

thermal plant are $k_{p2}$, $k_{i2}$, $k_{d2}$, $\lambda_{d2}$, and $\mu_2$. The source of contingency is a sudden change in the load in one area, which leads to high oscillations in the power system frequency and exchange power. This oscillation must be damped via LFC.

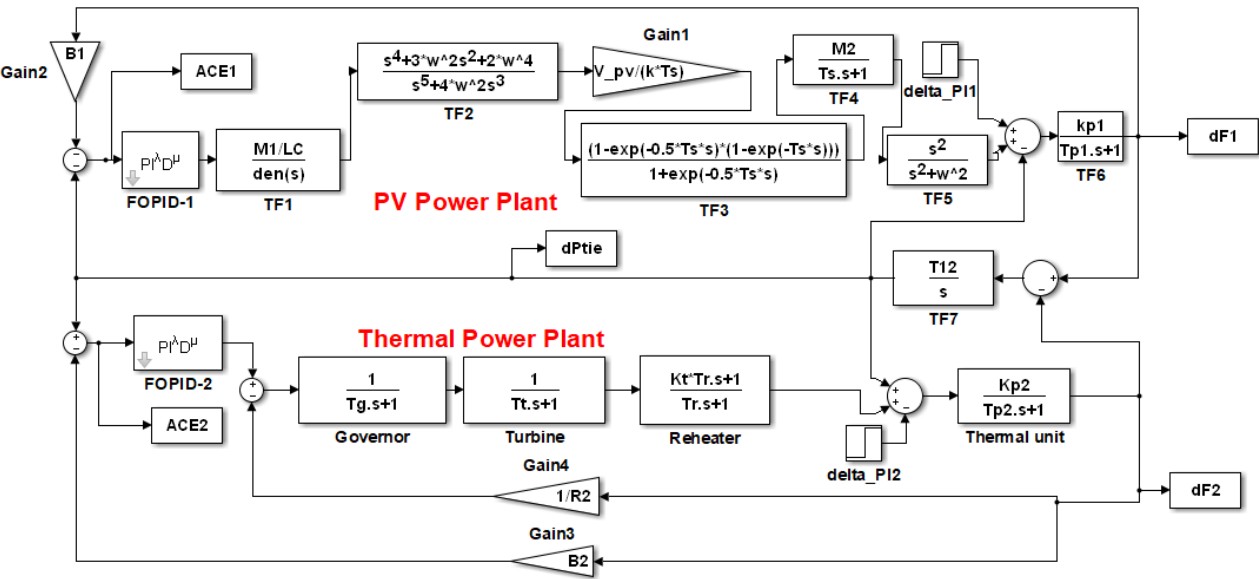

**Figure 7.** PV/thermal system in Simulink.

The first considered load disturbance is 10% on the PV plant. The optimal gains of the FOPID and the value of the best fitness function obtained via the proposed SOA and the others during $\Delta P_{L1} = 10\%$ are tabulated in Table 1. The results reveal that the best fitness value is 6.7506 obtained via the proposed SOA; the TDO comes in the second rank achieving ITAE of 6.7558. On the other hand, the worst optimizer is AOS, with a fitness value of 8.0539. Moreover, the elapsed time of each optimizer is measured. The proposed SOA is not the best in terms of computational time; however, it achieved the best fitness value which is the target of solving the considered problem. The variations of fitness value versus the iteration number for all considered approaches are shown in Figure 8. Moreover, the statistical parameters that assess the optimizer performance during the iteration process are given in Table 2. As the reader can see, the proposed SOA achieved the best variance and standard deviation (Std. dev.) with values of 1.1083 and 0.5277, respectively, outperforming all other optimizers. Furthermore, the time responses of $\Delta F_1$, $\Delta F_2$, and $\Delta P_{tie}$ for the PV/thermal system at $\Delta P_{L1} = 10\%$ are shown in Figure 9. The curves confirm the preference for the FOPID designed via the proposed SOA in such a case.

**Table 1.** FOPID parameters at $\Delta P_{L1} = 10\%$ in PV/thermal system.

|  | CBOA | JSOA | BO | TDO | AOS | SOA |
|---|---|---|---|---|---|---|
| $k_{p1}$ | 1.0000 | 1.0000 | 1.0000 | 0.99998 | 1.0000 | 1.0000 |
| $k_{i1}$ | 1.0000 | 1.0000 | 1.0000 | 0.48677 | 0.97191 | 0.9971 |
| $k_{d1}$ | 1.0000 | 1.0000 | 1.0000 | 1.0000 | 1.0000 | 1.0000 |
| $\lambda_{d1}$ | 0.7725 | 0.23273 | 0.64308 | 0.33975 | 0.98977 | 0.35946 |
| $\mu_1$ | 0.65477 | 1.0000 | 0.69726 | 0.83521 | 0.65565 | 0.87552 |
| $k_{p2}$ | 1.0000 | 1.0000 | 1.0000 | 0.26872 | 0.99882 | 0.99059 |
| $k_{i2}$ | 1.0000 | 1.0000 | 1.0000 | 0.99909 | 1.0000 | 0.8911 |
| $k_{d2}$ | 0.22547 | 1.0000 | 0.086092 | 0.99938 | 1.0000 | 0.13447 |
| $\lambda_{d2}$ | 0.53196 | 1.0000 | 0.00000 | 0.20817 | 1.0000 | 0.87058 |
| $\mu_2$ | 0.83065 | 1.0000 | 0.53543 | 0.77218 | 1.0000 | 0.977 |
| Elapsed time (Sec.) | 8924.533 | 3868.488 | 4265.285 | 6622.671 | 4707.0259 | 8151.158 |
| Fitness value | 7.1173 | 8.0539 | 6.9705 | 6.7558 | 8.0943 | 6.7506 |

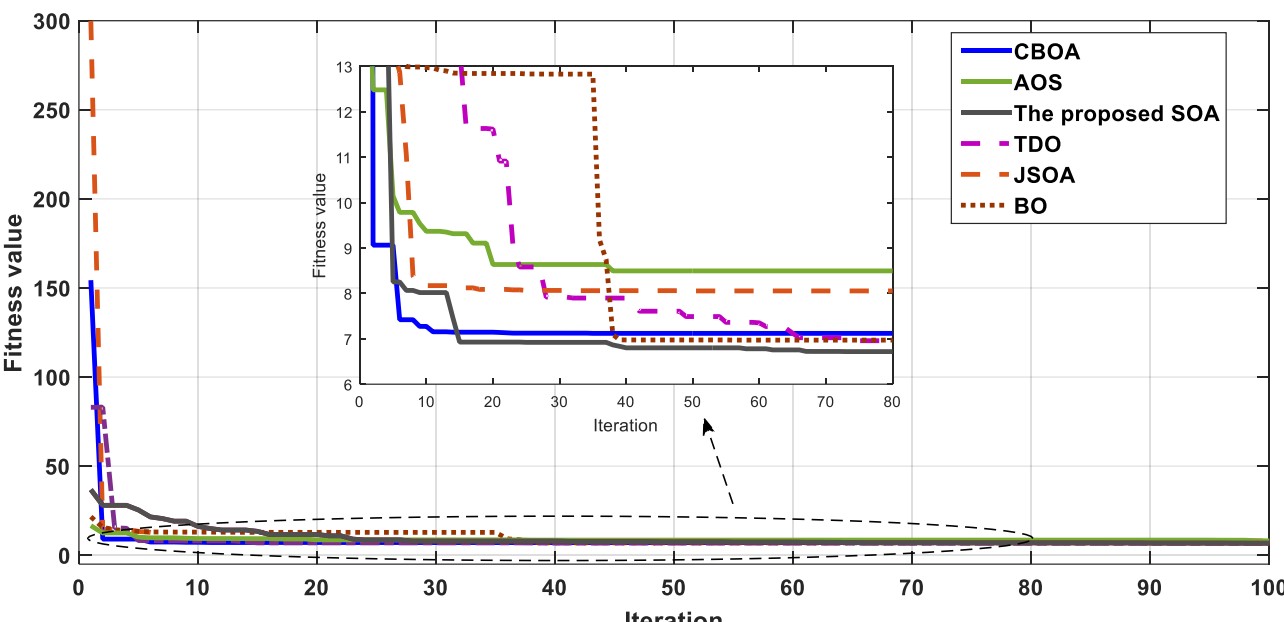

**Figure 8.** The fitness value versus the iteration number for PV/thermal system at $\Delta P_{L1}$ = 10%.

**Table 2.** Statistical parameters of the considered optimizers for PV/thermal system at $\Delta P_{L1}$ = 10%.

|          | CBOA   | JSOA   | BO     | TDO    | AOS    | SOA    |
| -------- | ------ | ------ | ------ | ------ | ------ | ------ |
| Best     | 7.1173 | 8.0539 | 6.9705 | 6.6985 | 8.0943 | 6.7506 |
| Worst    | 9.8586 | 16.552 | 98.501 | 9.2745 | 11.947 | 40.980 |
| Mean     | 7.9378 | 12.403 | 26.497 | 7.4199 | 9.8271 | 11.064 |
| Median   | 7.6988 | 12.107 | 10.046 | 7.3483 | 9.3298 | 7.7636 |
| Variance | 0.6422 | 16.461 | 994.99 | 0.5719 | 1.8280 | 1.1083 |
| Std. dev.| 0.8013 | 4.0572 | 31.544 | 0.7562 | 1.3520 | 0.5277 |

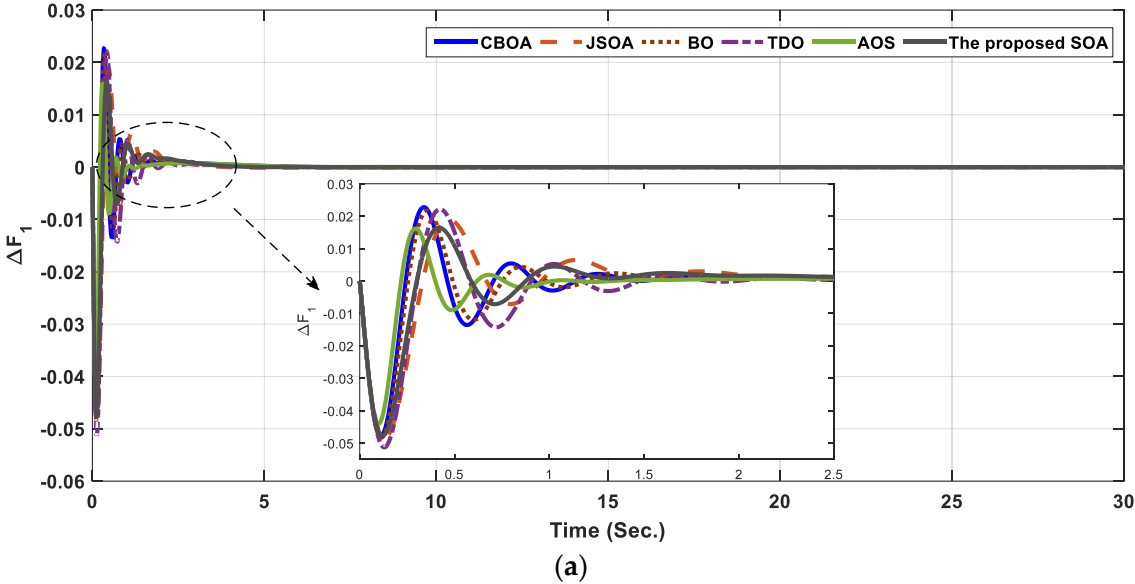

(**a**)

**Figure 9.** *Cont.*

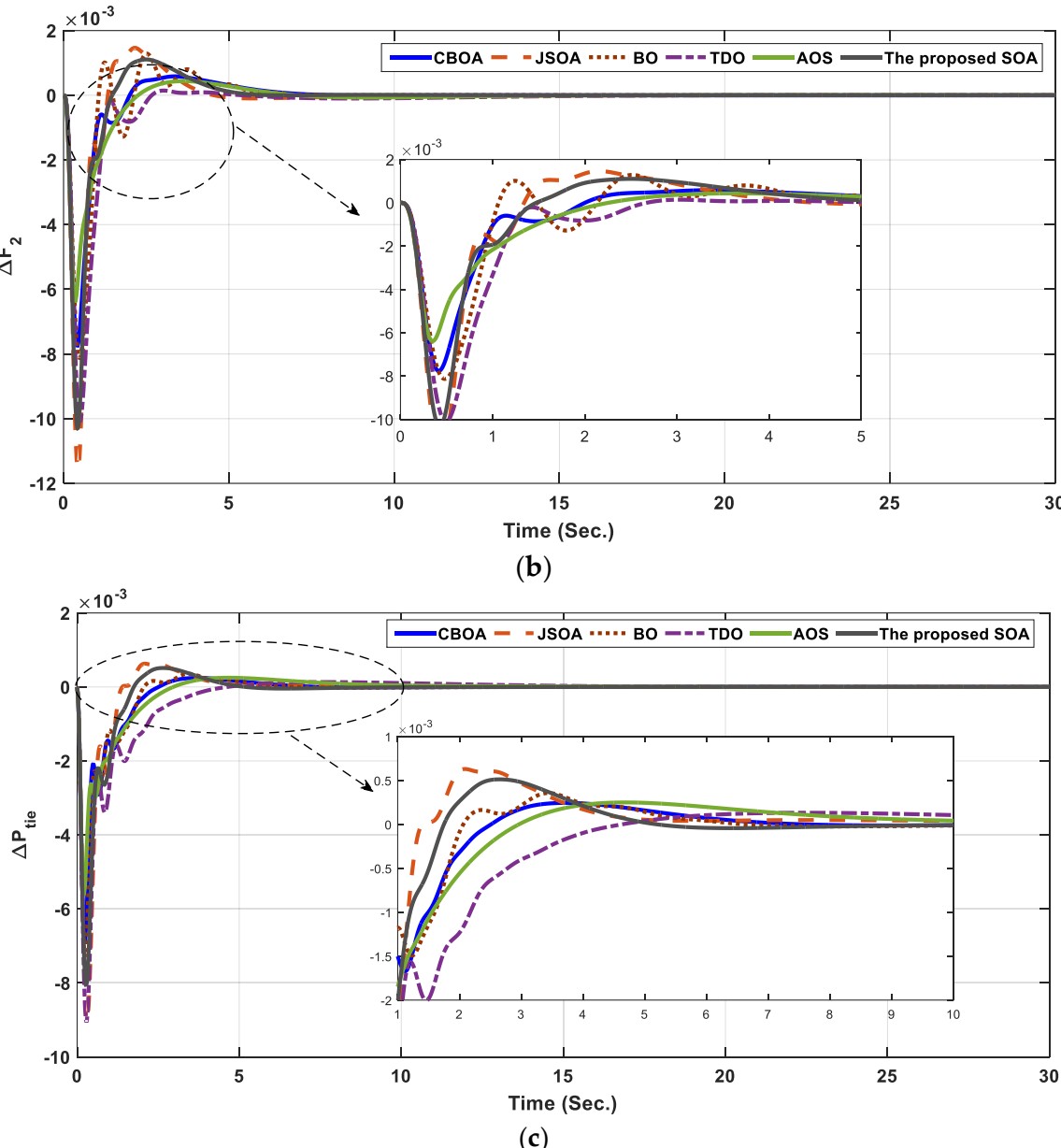

**Figure 9.** The time responses of (**a**) $\Delta F_1$, (**b**) $\Delta F_2$, and (**c**) $\Delta P_{tie}$ for PV/thermal system at $\Delta P_{L1} = 10\%$.

To confirm the proposed SOA validity, statistical tests were conducted using the Friedman ANOVA table, Wilcoxon rank test, Friedman rank test, and Kruskal Wallis test. The results of these tests are tabulated in Table 3. The Wilcoxon test results revealed that, JSOA and AOS reject the null equal medians while CBOA, BO, and TDO accept the null equal medians. Therefore, the SOA has a significant difference from the other approaches. Regarding the Friedman rank test, the SOA came in the first rank with a value of 3.2, then CBOA, TDO, BO, AOS, and JSOA came next to the proposed approach. The *p*-values for the Friedman, ANOVA, and Kruskal Wallis tests were 6.8140e-05, 0.0308, and 1.3583e-04, respectively. The Friedman test *p*-value confirmed the significant difference in column means, while Kruskal Wallis *p*-value revealed the rejection of the null hypothesis of data with the same distribution. The fitness functions during separate trials via ANOVA Wallis for the PV/thermal system at $\Delta P_{L1} = 10\%$ are shown in Figure 10. Regarding the boxplots given in Figure 10, the penguins obtained via the proposed SOA, TDO, and CBOA have the smallest flippers, while the others have larger flippers. This confirms that CBOA, TDO, and the proposed SOA reject the null hypothesis.

**Table 3.** Statistical tests of Friedman ANOVA, Wilcoxon rank, Friedman rank, and Kruskal Wallis for PV/thermal system at $\Delta P_{L1}$ = 10%.

|  |  | CBOA | JSOA | BO | TDO | AOS | SOA |
|---|---|---|---|---|---|---|---|
| Wilcoxon rank test | *p*-value | 0.7913 | 0.0090 | 0.1405 | 0.1212 | 0.0073 | - |
|  | h-value | 0 | 1 | 0 | 0 | 1 | - |
| Null hypothesis rejection |  | × | √ | × | × | √ | - |
| Friedman rank |  | 4.4 | 9.6 | 8.1 | 5.0 | 8.7 | 3.2 |
| Friedman test *p*-value |  | 6.8140e-05 |  |  |  |  |  |
| *p*-value based on ANOVA |  | 0.0308 |  |  |  |  |  |
| *p*-value based on Kruskal Wallis test |  | 1.3583e-04 |  |  |  |  |  |

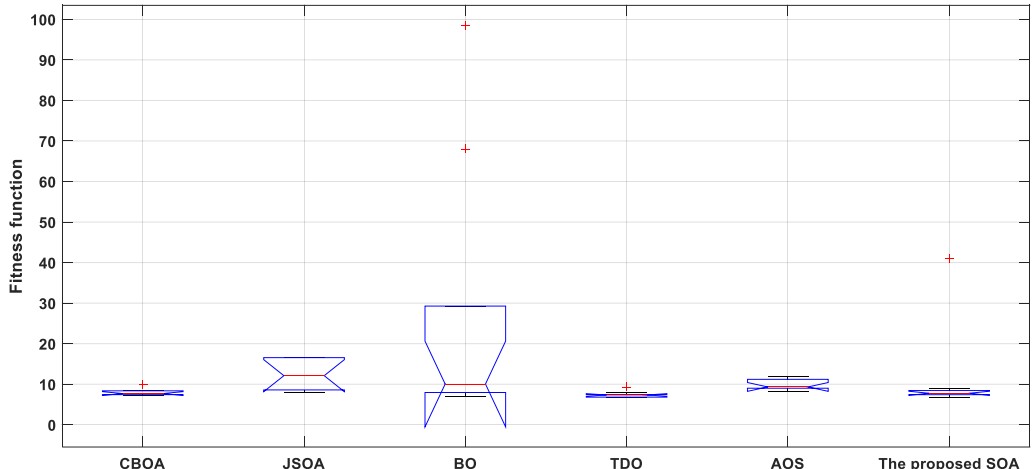

**Figure 10.** The fitness functions during separate trials via ANOVA Wallis for PV/thermal system at $\Delta P_{L1}$ = 10%.

A second load disturbance of 10% is assumed at the thermal plant: the optimal parameters of the controllers via SOA and others are given in Table 4 in addition to the other approaches. The proposed SOA succeeded in achieving the best ITAE of 1.8779, outperforming all considered approaches. The worst approach is JSOA, with a fitness value of 3.8959.

**Table 4.** FOPID parameters at $\Delta P_{L2}$ = 10% in PV/thermal system.

|  | CBOA | JSOA | BO | TDO | AOS | SOA |
|---|---|---|---|---|---|---|
| $k_{p1}$ | 0.36569 | 0.1000 | 0.1000 | 0.10003 | 0.11668 | 0.1001 |
| $k_{i1}$ | 0.1 | 1.0000 | 0.1000 | 0.10485 | 0.10334 | 0.1 |
| $k_{d1}$ | 0.3708 | 0.1000 | 0.71061 | 0.99984 | 0.11471 | 0.14113 |
| $\lambda_{d1}$ | 0.8262 | 1.0000 | 0.61687 | 0.67797 | 0.72813 | 0.20952 |
| $\mu_1$ | 0.92171 | 0.90165 | 0.99761 | 0.94303 | 0.96683 | 0.78574 |
| $k_{p2}$ | 0.87133 | 1.0000 | 0.99612 | 0.20311 | 0.11802 | 0.39052 |
| $k_{i2}$ | 1.0000 | 1.0000 | 1.0000 | 0.99984 | 0.99783 | 1.0000 |
| $k_{d2}$ | 1.0000 | 1.0000 | 1.0000 | 0.99577 | 0.96641 | 1.0000 |
| $\lambda_{d2}$ | 0.28196 | 1.0000 | 0.2517 | 0.10308 | 0.96431 | 0.28383 |
| $\mu_2$ | 0.95274 | 0.1000 | 0.99088 | 0.87324 | 0.11028 | 0.73404 |
| Fitness value | 2.9092 | 3.8959 | 2.0181 | 2.0489 | 2.4266 | 1.8779 |

The performances of the approaches during the iterative process are shown in Figure 11; the curves confirmed the preference for the proposed SOA. The time responses of frequency and power exchange are given in Figure 12. The proposed SOA-based methodology achieved excellent performance for the PV/thermal interconnected power system subjected to different load disturbances.

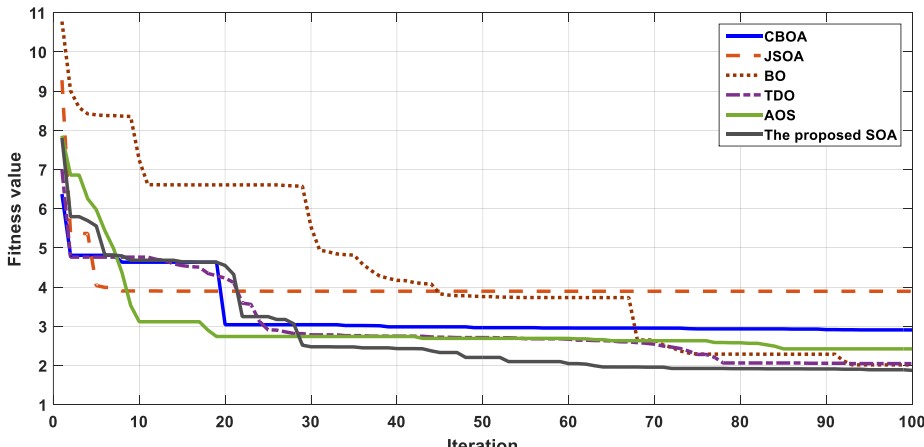

**Figure 11.** The fitness value versus the iteration number for PV/thermal system at $\Delta P_{L2} = 10\%$.

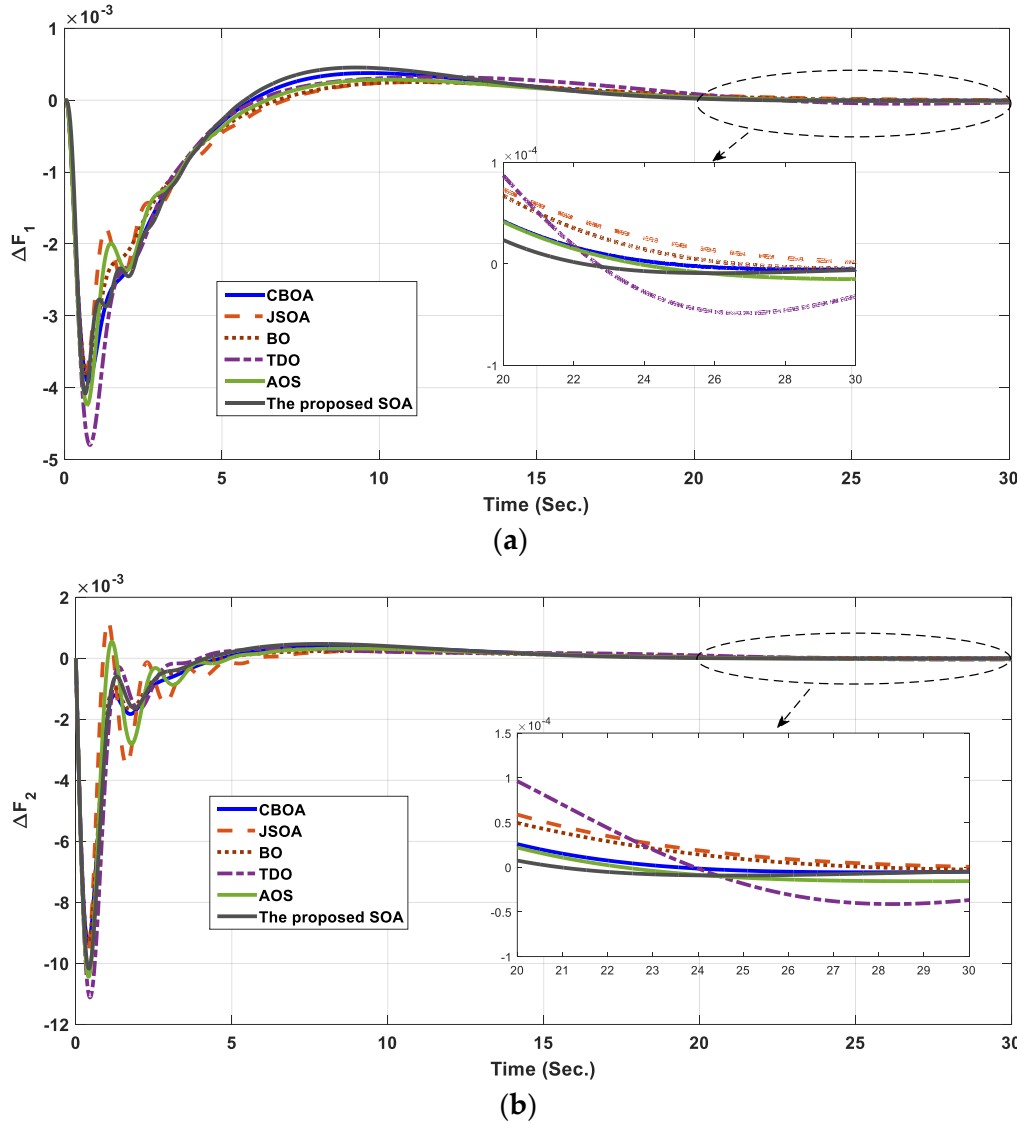

**Figure 12.** *Cont.*

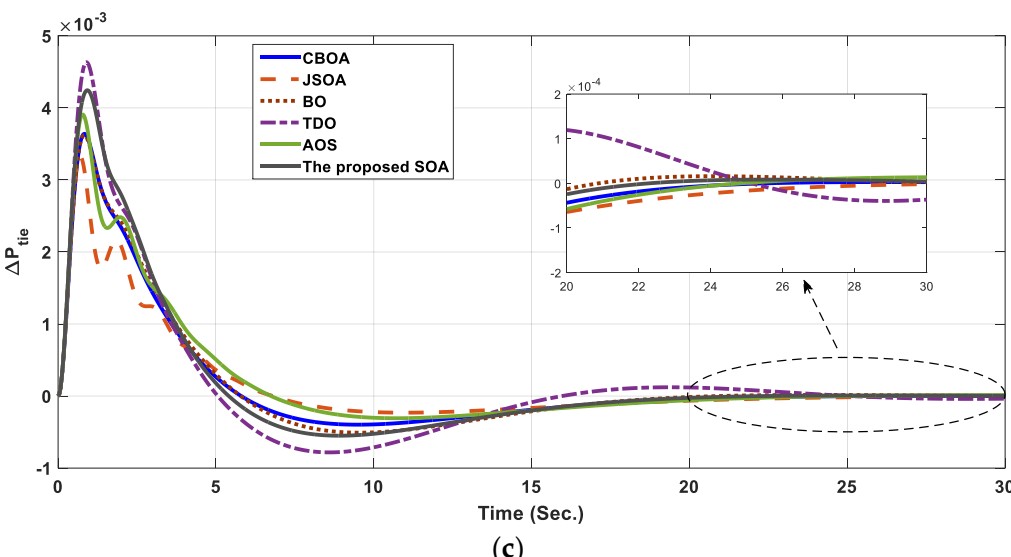

**Figure 12.** The time responses of (**a**) $\Delta F_1$, (**b**) $\Delta F_2$, and (**c**) $\Delta P_{tie}$ for PV/thermal system at $\Delta P_{L2} = 10\%$.

## 6.2. Four Interconnected Power System

Most power systems have many generating units, which may be conventional and/or renewable energy. In this section, the authors analyze the proposed LFC-FOPID with a thermal/WT/thermal/PV system; the GDB and GRC of thermal units are considered (see Figure 4). This configuration is constructed in Simulink/Matlab, as shown in Figure 13. In such a system, four FOPID controllers are installed with the four considered plants. The controller with the first thermal plant has adapted parameters of $k_{p1}$, $k_{i1}$, $k_{d1}$, $\lambda_{d1}$, and $\mu_{1,}$ while the second with the wind energy plant (Area 2) has $k_{p2}$, $k_{i2}$, $k_{d2}$, $\lambda_{d2}$, and $\mu_2$. The third controller is installed with the second thermal plant (Area 3), it has adapted parameters of $k_{p3}$, $k_{i3}$, $k_{d3}$, $\lambda_{d3}$, and $\mu_3$. The last controller is installed with the PV plant (Area 4) with parameters of $k_{p4}$, $k_{i4}$, $k_{d4}$, $\lambda_{d4}$, and $\mu_4$. These parameters are identified via the proposed SOA. The first considered disturbance is 1% on the first thermal plant; the results obtained in such a case are tabulated in Table 5. The obtained results revealed that, the proposed SOA achieved the first rank in term of fitness value with a value of 0.0327, TDO comes in the second rank with a fitness value of 0.0368, while the worst optimizer is BO with ITAE of 0.3565. Moreover, the time responses of frequency and exchange power violations obtained via all considered approaches are shown in Figure 14; the zoomed curves show the better performance of SOA compared to the others. The time response performance specifications of frequency and tie-line power violations, including rise time ($t_r$), settling time ($t_s$), minimum settling time ($t_{s,min}$), maximum settling time ($t_{s,max}$), overshoot (OS), undershoot (Us), and peak time ($t_p$) are calculated and given in Table 6. These parameters are helpful in clarifying the preference for the proposed SOA in achieving system stability after an acceptable time.

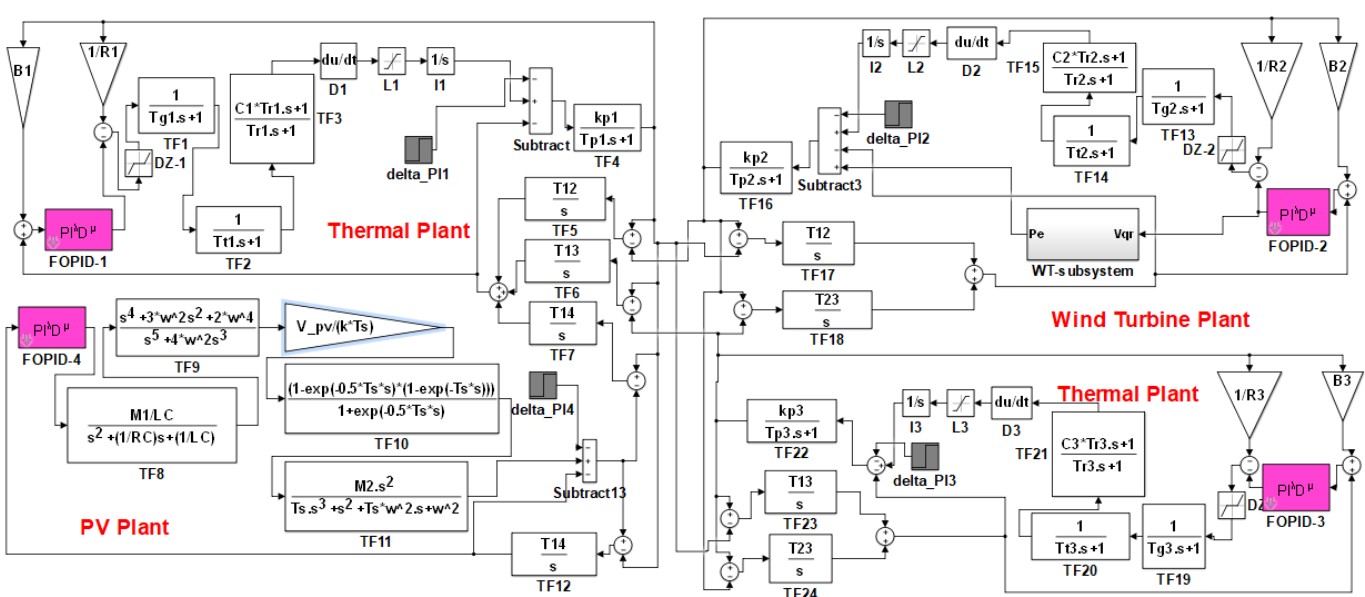

**Figure 13.** Simulink model of thermal/WT/thermal/PV interconnected system.

**Table 5.** FOPID parameters at $\Delta P_{L1}$ = 1% in thermal/WT/thermal/PV system.

|  | CBOA | JSOA | BO | TDO | AOS | SOA |
|---|---|---|---|---|---|---|
| $k_{p1}$ | 0.030335 | 0.015797 | 1.0000 | 0.01 | 0.013113 | 0.027242 |
| $k_{i1}$ | 0.013487 | 0.020344 | 0.010001 | 0.102469 | 0.150434 | 0.014646 |
| $k_{d1}$ | 0.01 | 0.01 | 0.0100 | 0.01 | 0.033762 | 0.032416 |
| $\lambda_{d1}$ | 0.016799 | 0.01 | 0.999969 | 0.010007 | 0.0100 | 0.031366 |
| $\mu_1$ | 0.033807 | 0.01 | 0.010001 | 0.01 | 0.0100 | 0.03194 |
| $k_{p2}$ | 0.054376 | 0.01 | 0.278846 | 0.010621 | 0.059971 | 0.024249 |
| $k_{i2}$ | 0.014237 | 0.01 | 0.253138 | 0.029362 | 0.0100 | 0.032453 |
| $k_{d2}$ | 0.010332 | 0.064301 | 0.066672 | 0.01001 | 0.0100 | 0.023304 |
| $\lambda_{d2}$ | 0.014425 | 0.01 | 0.482488 | 0.493617 | 0.025474 | 0.012931 |
| $\mu_2$ | 0.032054 | 0.01 | 0.0100 | 0.119511 | 0.02015 | 0.030971 |
| $k_{p3}$ | 0.023032 | 0.01 | 0.834948 | 0.018221 | 0.05298 | 0.012431 |
| $k_{i3}$ | 0.025541 | 0.016533 | 0.858446 | 0.010075 | 0.085947 | 0.025044 |
| $k_{d3}$ | 0.088584 | 0.065911 | 1.0000 | 0.170223 | 0.034121 | 0.031075 |
| $\lambda_{d3}$ | 0.01 | 0.106448 | 0.642914 | 0.167097 | 0.0100 | 0.030966 |
| $\mu_3$ | 0.042053 | 0.012318 | 0.444737 | 0.01 | 0.32664 | 0.012818 |
| $k_{p4}$ | 0.01 | 0.02 | 0.0100 | 0.036939 | 0.012773 | 0.018763 |
| $k_{i4}$ | 0.01134 | 0.015331 | 0.020992 | 0.010394 | 0.032627 | 0.019244 |
| $k_{d4}$ | 0.028713 | 0.01 | 0.47806 | 0.01 | 0.011926 | 0.015534 |
| $\lambda_{d4}$ | 0.076208 | 0.028389 | 0.0100 | 0.012869 | 0.010865 | 0.0312 |
| $\mu_4$ | 0.01 | 0.018096 | 1.000 | 0.098181 | 0.253268 | 0.032036 |
| Elapsed time (Sec.) | 10,889.171 | 6675.141 | 6287.736 | 10,042.1075 | 6630.9976 | 9471.469 |
| Fitness value | 0.0511 | 0.06301 | 0.3565 | 0.0368 | 0.0571 | 0.0327 |

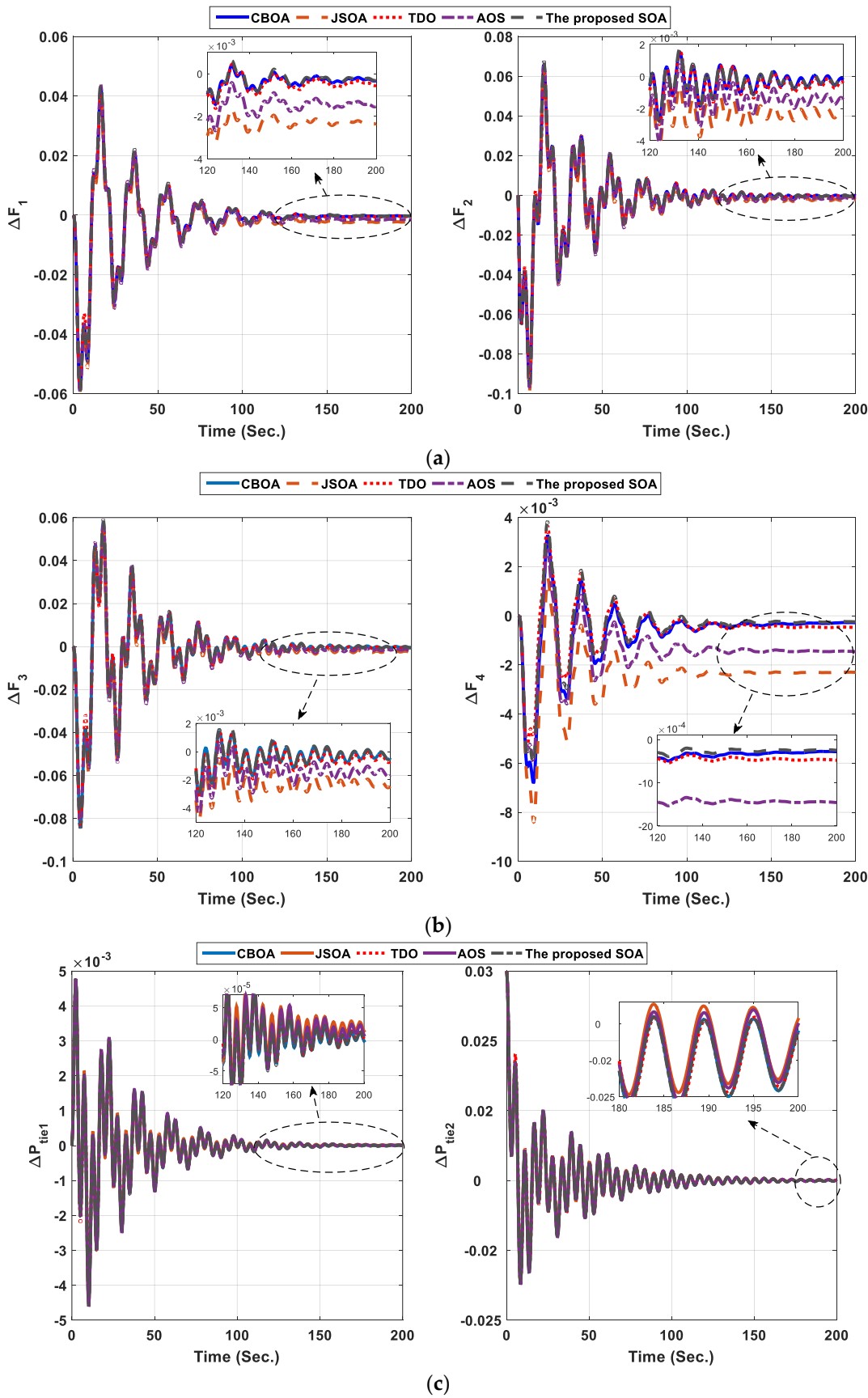

**Figure 14.** *Cont.*

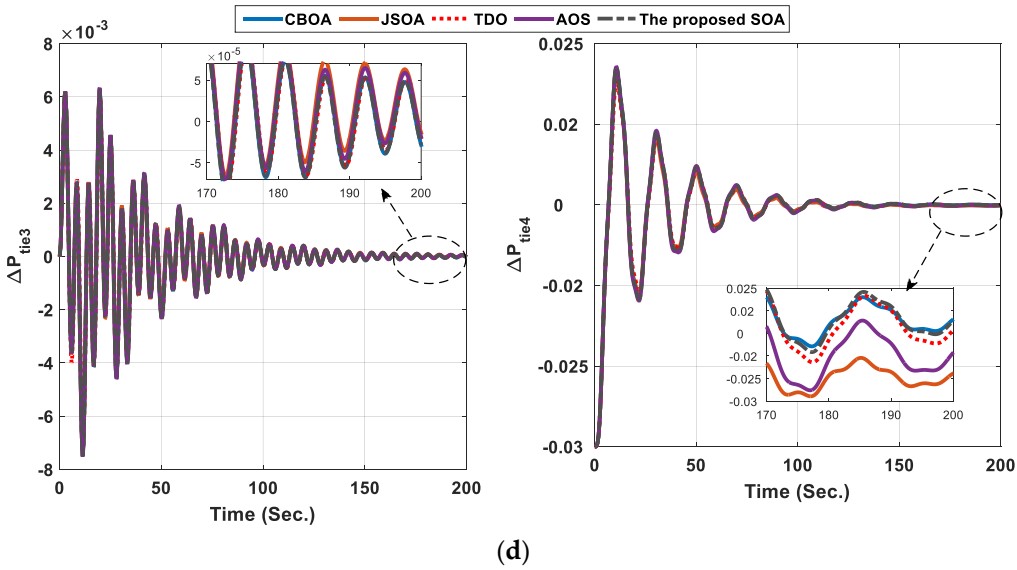

(**d**)

**Figure 14.** The time responses of (**a**) $\Delta F_1$, $\Delta F_2$ (**b**) $\Delta F_3$, $\Delta F_4$, (**c**) $\Delta P_{tie1}$, $\Delta P_{tie2}$, and (**d**) $\Delta P_{tie3}$, $\Delta P_{tie4}$ of thermal/WT/thermal/PV system at $\Delta P_{L1} = 1\%$.

**Table 6.** Performance specifications of $\Delta F_1$, $\Delta F_2$, $\Delta F_3$, $\Delta F_4$, $\Delta P_{tie1}$, $\Delta P_{tie2}$, $\Delta P_{tie3}$, and $\Delta P_{tie4}$ of four interconnected system at $\Delta P_{L1} = 1\%$.

| | | $t_r$ (Sec.) | $t_s$ (Sec.) | $t_{s,min}$ (Sec.) | $t_{s,max}$ (Sec.) | Os (pu) | Us (pu) | $t_p$ (Sec.) |
|---|---|---|---|---|---|---|---|---|
| $\Delta F_1$ | CBOA | 0.218126 | 112.4181 | −0.05914 | 0.042865 | 17530.68 | 12779.44 | 3.967595 |
| | JSOA | 0.42883 | 112.3608 | −0.05925 | 0.041337 | 2430.683 | 1765.517 | 3.976048 |
| | BO | 5.394888 | 99.65318 | −0.04436 | 0.064188 | 533.3823 | 782.7876 | 6.314557 |
| | TDO | 0.259345 | 116.7282 | −0.05873 | 0.040585 | 10456.03 | 7294.803 | 3.943959 |
| | AOS | 0.369491 | 116.4693 | −0.05909 | 0.043968 | 3735.791 | 2854.158 | 3.967459 |
| | SOA | 0.213651 | 116.1438 | −0.05913 | 0.044413 | 18632.35 | 14071.23 | 3.966584 |
| | | $t_r$ (Sec.) | $t_s$ (Sec.) | $t_{s,min}$ (Sec.) | $t_{s,max}$ (Sec.) | Os (pu) | Us (pu) | $t_p$ (Sec.) |
| $\Delta F_2$ | CBOA | 0.002726 | 124.7015 | −0.09662 | 0.065934 | 47160.75 | 32251.03 | 6.758115 |
| | JSOA | 0.029639 | 124.4023 | −0.09886 | 0.064269 | 4352.023 | 2894.324 | 6.707654 |
| | BO | 0.004961 | 99.71404 | −0.04461 | 0.077011 | 37488.43 | 37156.86 | 73.48295 |
| | TDO | 0.005394 | 133.0857 | −0.08998 | 0.063528 | 22146.13 | 15705.53 | 6.754989 |
| | AOS | 0.018928 | 132.897 | −0.09696 | 0.067587 | 6734.498 | 4764.106 | 6.739581 |
| | SOA | 0.002528 | 132.5271 | −0.09692 | 0.067685 | 51019.06 | 35699.57 | 6.745544 |
| | | $t_r$ (Sec.) | $t_s$ (Sec.) | $t_{s,min}$ (Sec.) | $t_{s,max}$ (Sec.) | Os (pu) | Us (pu) | $t_p$ (Sec.) |
| $\Delta F_3$ | CBOA | 0.243729 | 135.2427 | −0.08438 | 0.057852 | 17936.97 | 12365.7 | 4.385474 |
| | JSOA | 0.432962 | 134.8931 | −0.08456 | 0.05576 | 3334.456 | 2264.804 | 4.394015 |
| | BO | 0.661445 | 99.91756 | −0.06307 | 0.087805 | 744.1215 | 1175.131 | 76.12008 |
| | TDO | 0.281637 | 135.799 | −0.08369 | 0.056156 | 11618.06 | 7862.521 | 4.354946 |
| | AOS | 0.378296 | 135.5671 | −0.0844 | 0.058481 | 4932.842 | 3487.269 | 4.386527 |
| | SOA | 0.240396 | 135.5081 | −0.08441 | 0.059358 | 18688.96 | 13212.79 | 4.385127 |
| | | $t_r$ (Sec.) | $t_s$ (Sec.) | $t_{s,min}$ (Sec.) | $t_{s,max}$ (Sec.) | Os (pu) | Us (pu) | $t_p$ (Sec.) |
| $\Delta F_4$ | CBOA | 0.646564 | 129.2411 | −0.00681 | 0.003278 | 2336.508 | 1173.331 | 9.174419 |
| | JSOA | 1.339719 | 125.0202 | −0.00841 | 0.001541 | 265.737 | 67.00468 | 9.349339 |
| | BO | 77.84681 | 99.29976 | 0.013333 | 0.019746 | 21.57953 | 28.14631 | 84.9515 |
| | TDO | 0.772649 | 137.297 | −0.00538 | 0.003661 | 1025.934 | 766.6746 | 5.254582 |
| | AOS | 1.150486 | 136.6 | −0.00596 | 0.002898 | 308.7192 | 198.8507 | 9.114361 |
| | SOA | 0.625838 | 126.1221 | −0.00607 | 0.00385 | 2357.405 | 1559.14 | 9.069103 |

**Table 6.** *Cont.*

|  |  | $t_r$ (Sec.) | $t_s$ (Sec.) | $t_{s,min}$ (Sec.) | $t_{s,max}$ (Sec.) | Os (pu) | Us (pu) | $t_p$ (Sec.) |
|---|---|---|---|---|---|---|---|---|
| $\Delta P_{tie-1}$ | CBOA | 0.000637 | 118.5854 | −0.00452 | 0.003022 | 161006.8 | 170204.2 | 2.364085 |
|  | JSOA | 0.049687 | 118.455 | −0.0046 | 0.004776 | 36540.76 | 35302.54 | 2.36612 |
|  | BO | 0.882234 | 99.87703 | −0.01834 | −0.00146 | 104.4635 | 0 | 7.694692 |
|  | TDO | 0.013741 | 118.7534 | −0.00409 | 0.004767 | 505700.3 | 433561.9 | 2.360746 |
|  | AOS | 0.035581 | 125.6115 | −0.00461 | 0.004773 | 71280.16 | 68873.31 | 2.36558 |
|  | SOA | 0.000582 | 118.6747 | −0.00459 | 0.003083 | 178819.4 | 186219 | 2.366509 |
|  |  | $t_r$ (Sec.) | $t_s$ (Sec.) | $t_{s,min}$ (Sec.) | $t_{s,max}$ (Sec.) | Os (pu) | Us (pu) | $t_p$ (Sec.) |
| $\Delta P_{tie-2}$ | CBOA | 6.057111 | 125.7488 | −0.02237 | −0.01003 | 49.54956 | 0 | 8.337896 |
|  | JSOA | 6.007973 | 125.6896 | −0.0224 | −0.01006 | 49.87361 | 0 | 8.2875 |
|  | BO | 1.163071 | 99.60019 | −0.01051 | 0.011322 | 179.7168 | 259.602 | 80.0753 |
|  | TDO | 6.183642 | 125.8925 | −0.02206 | −0.01013 | 47.45189 | 0 | 8.469989 |
|  | AOS | 6.040323 | 125.8636 | −0.02244 | −0.00994 | 50.11094 | 0 | 8.328108 |
|  | SOA | 6.046111 | 125.7889 | −0.02242 | −0.00996 | 49.86594 | 0 | 8.331606 |
|  |  | $t_r$ (Sec.) | $t_s$ (Sec.) | $t_{s,min}$ (Sec.) | $t_{s,max}$ (Sec.) | Os (pu) | Us (pu) | $t_p$ (Sec.) |
| $\Delta P_{tie-3}$ | CBOA | 0.004796 | 159.2212 | −0.00745 | 0.00622 | 26006.25 | 21791.22 | 11.3151 |
|  | JSOA | 0.002346 | 159.1255 | −0.00744 | 0.00625 | 53840.32 | 45316.67 | 11.271 |
|  | BO | 1.105488 | 99.86355 | −0.00686 | 0.005781 | 50.67886 | 178.6869 | 9.799218 |
|  | TDO | 0.003439 | 159.3877 | −0.00722 | 0.006008 | 33852.81 | 29085.48 | 11.43047 |
|  | AOS | 0.003184 | 159.2393 | −0.00752 | 0.00634 | 39593.22 | 33481.55 | 11.30361 |
|  | SOA | 0.004757 | 159.2366 | −0.00751 | 0.006278 | 26456.8 | 22202.1 | 11.30935 |
|  |  | $t_r$ (Sec.) | $t_s$ (Sec.) | $t_{s,min}$ (Sec.) | $t_{s,max}$ (Sec.) | Os (pu) | Us (pu) | $t_p$ (Sec.) |
| $\Delta P_{tie-4}$ | CBOA | 3.811909 | 99.87232 | 0.00929 | 0.02338 | 55.94675 | 0 | 10.6354 |
|  | JSOA | 3.795319 | 99.11462 | 0.00931 | 0.023423 | 56.73186 | 0 | 10.57403 |
|  | BO | 0.472669 | 98.00973 | −0.00196 | 0.024023 | 2113.074 | 180.1916 | 12.07693 |
|  | TDO | 3.871557 | 110.1162 | 0.009571 | 0.022714 | 51.59852 | 0 | 10.77417 |
|  | AOS | 3.765408 | 109.9693 | 0.009032 | 0.023574 | 57.54973 | 0 | 10.62046 |
|  | SOA | 3.789229 | 100.6543 | 0.009114 | 0.023548 | 57.07933 | 0 | 10.63256 |

It is essential to investigate the designed FOPID controller via the proposed SOA during variable disturbances. Therefore, the second disturbance considered in the four interconnected power systems is variable step one, as shown in Figure 15. The time responses of the frequency and tie-line power violations for this case are shown in Figure 16. The proposed FOPID controllers succeeded in achieving a stable system by banishing the changes in frequencies and exchange powers occurring during that disturbance.

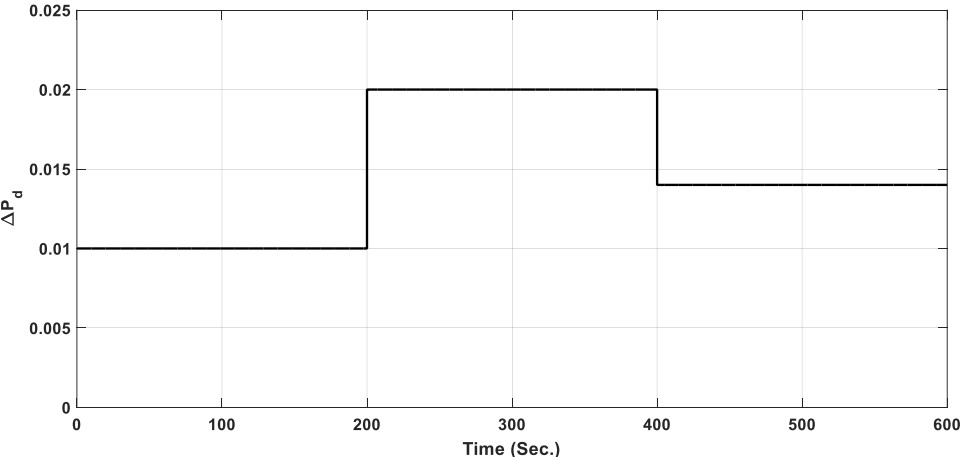

**Figure 15.** Pattern of variable disturbance.

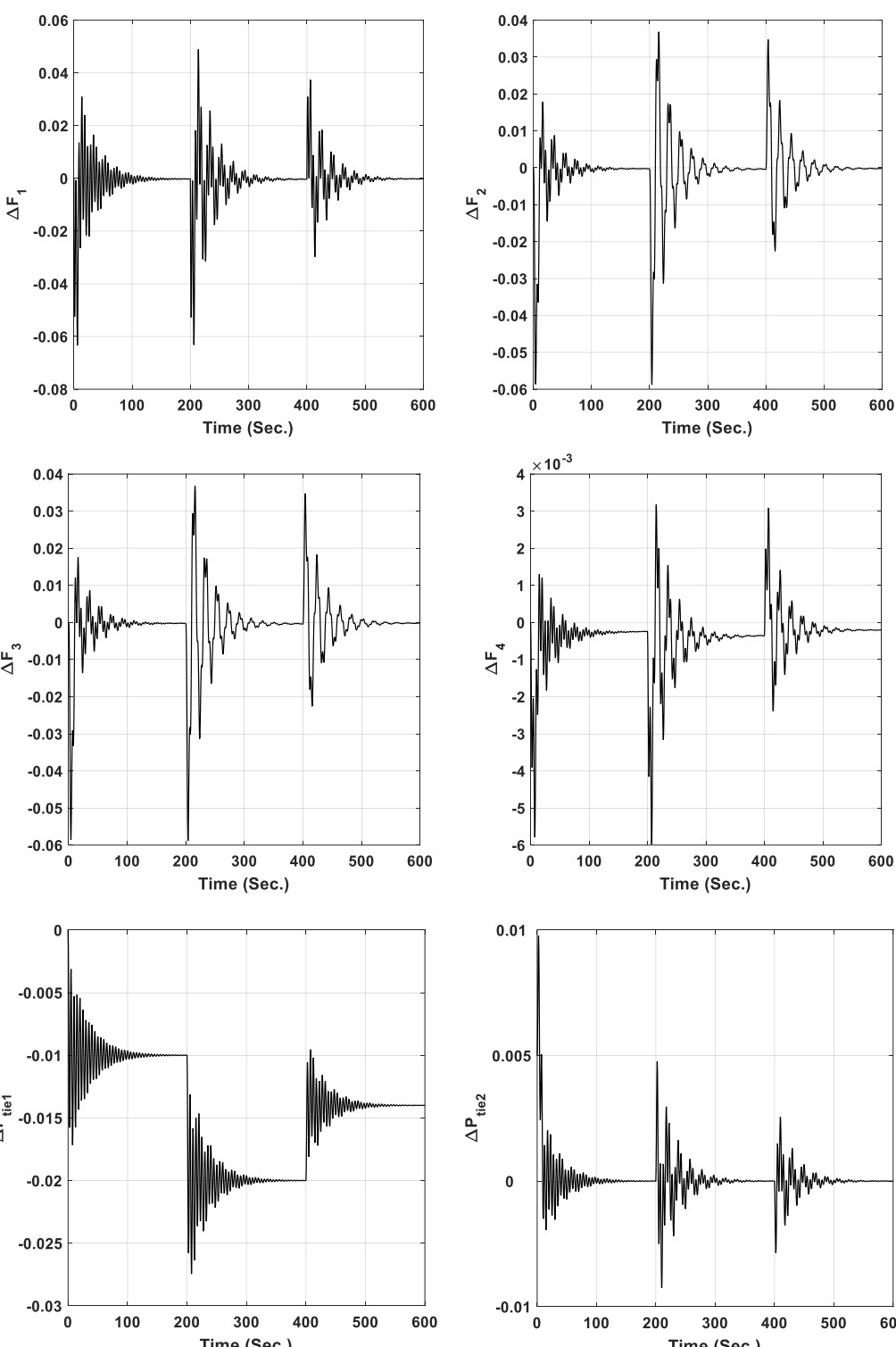

**Figure 16.** *Cont.*

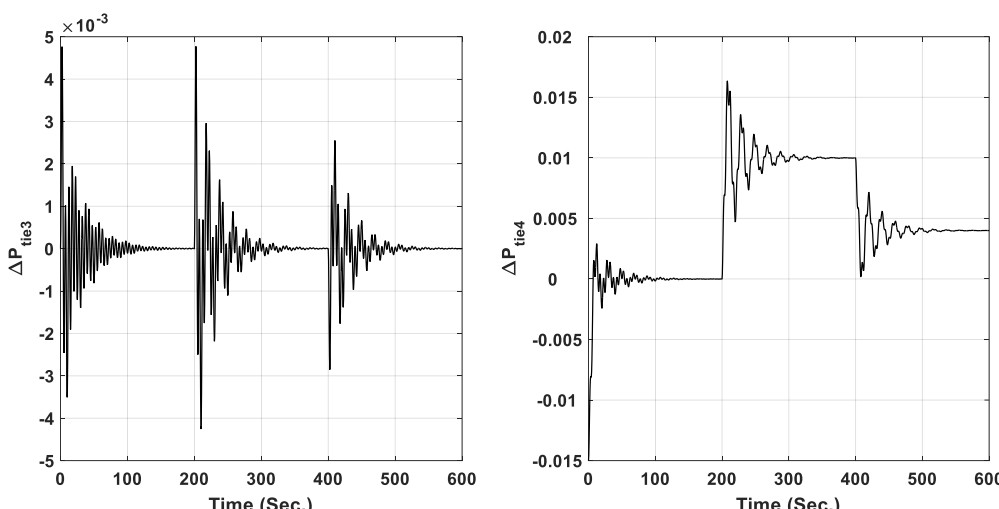

**Figure 16.** The time responses of frequency and exchange power violations for thermal/WT/thermal/PV system with variable disturbance in area 1.

Finally, the authors recommend the SOA as an efficient tool to design FOPID-LFC installed in multi-interconnected multi-sources power systems, as the obtained results confirmed its preferability in all studied load disturbances.

## 7. Conclusions

This work introduces a recent skill optimization algorithm to identify the optimal parameters of fractional-order proportional integral derivative load frequency control installed with an interconnected system, including renewable energy sources. The target is to mitigate the integral time absolute error of frequency and exchange power violations. Two-area and four-area power systems are analyzed. The first system has photovoltaic and thermal plants, while the second one comprises photovoltaic, wind turbine, and two thermal plants with governor dead-band and generation rate constraints. In the photovoltaic/thermal system, two disturbances are analyzed, which are 10% on the first plant and the same on the thermal plant. In the thermal/wind turbine/thermal/photovoltaic system, 1% and variable load disturbances on the first thermal area are simulated. The proposed algorithm is compared to chef-based optimization algorithm, jumping spider optimization algorithm, Bonobo optimization, Tasmanian devil optimization, and Atomic orbital search. The authors also conducted statistical tests (Friedman ANOVA table, Wilcoxon rank test, Friedman rank test, and Kruskal Wallis test) to assess the proposed approach. The best fitness values are obtained via the proposed approach with values of 1.8779 pu and 0.0327 pu for the two and four areas models, respectively. The capability and robustness of the proposed controller designed via the proposed approach are confirmed. Accelerating the response of the designed controllers in a multi-interconnected area via a hybrid optimization approach is recommended for future work. Moreover, investigating the stability of the system will be considered in the next paper.

**Author Contributions:** Data curation, A.F. and H.R.; Formal analysis, S.F.; Methodology, R.M.G. and R.A.; Software, M.M.G. All authors have read and agreed to the published version of the manuscript.

**Funding:** This work was funded by the Deanship of Scientific Research at Princess Nourah bint Abdulrahman University, through the Research Groups Program Grant no. (RGP-1443-0046).

**Institutional Review Board Statement:** Not applicable.

**Informed Consent Statement:** Not applicable.

**Data Availability Statement:** Not applicable.

**Conflicts of Interest:** The authors declare no conflict of interest.

## Nomenclature

| | | | | |
|---|---|---|---|---|
| $I$ | PV panel output current | | $r$ | Radius of turbine |
| $V_{pv}$ | PV panel output voltage | | $V_w$ | Wind speed |
| $n_s$ | Number of series cells | | $\rho$ | Air density |
| $n_p$ | Number of parallel cells | | $A$ | Swept area of turbine blades |
| $\varepsilon$ | Factor of completion | | $G_t$ | Steam turbine transfer function |
| $k$ | Boltzmann constant | | $G_g$ | Governor transfer function |
| $T$ | PV panel temperature | | $G_r$ | Reheater transfer function |
| $Q$ | Electron charge | | $G_{gen}$ | Generator transfer function |
| $G$ | Irradiance in W/m$^2$ | | $K_t$ | Steam turbine gain |
| $I_{ph}$ | Photo current | | $K_g$ | Governor gain |
| $I_o$ | Saturation current | | $K_r$ | Reheater gain |
| $R_s$ | Cell series resistance | | $K_p$ | Generator gain |
| $P_{pv}$ | PV module output power | | $T_t$ | Steam turbine time constant |
| $\omega$ | Angular frequency of grid | | $T_g$ | Governor time constant |
| $R$ | Converter output resistance | | $T_r$ | Reheater time constant |
| $C$ | Converter output capacitance | | $T_p$ | Generator time constant |
| $L$ | Converter output inductance | | $k_p, k_i, k_d, \lambda_d,$ and $\mu$ | Parameters of FOPID controller |
| $T_s$ | Simulation time | | $K_{pw1}, K_{pw2},$ and $K_{pw3}$ | Wind plant gains |
| $M_1$ | Buck converter voltage gain | | $T_{pw1}, T_{pw2},$ and $T_{pw3}$ | Wind plant time constants |
| $M_2$ | Inverter converter voltage gain | | $e(t)$ | FOPID controller input |
| $C_p$ | Power coefficient | | $\Delta F_i$ | Violations in $i$th area frequency |
| $\lambda$ | Tip ratio | | $P_{tie,i}$ | Violations in $i$th area exchange power |
| $\beta$ | Pitch angle of blade | | $t$ | Specified time |
| $\omega_t$ | Mechanical angular speed of turbine | | $n_a$ | Number of interconnected plants |

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
