# Peer review of "A New Fractional-Order Load Frequency Control for Multi-Renewable Energy Interconnected Plants Using Skill Optimization Algorithm"

_sustainability, doi:10.3390/su142214999_

Round 1

Reviewer 1 Report

This paper proposes a heuristic-based optimization approach to develop load frequency controller to manage frequency constraints during contingency issues in interconnected power systems. The performance of the proposed approach has been compared with other approaches existing in the literature. The presentation of the paper is not adequate. In particular, detailed clarifications and descriptions   of the adopted models, controllers and parameters are needed. This reviewer has the additional following comments that have to be adequately addressed/discussed:

-          The novelty of this work has to be better illustrated in the Introduction. The description of the proposed approach is also missing in the Introduction.

-          Please add the outline of the paper at the end of the Introduction Section.

-          The variables in Eq.3 and Eq. 4 are not all defined.

-          Please provide further details about the adopted parameters in the controller presented in Fig. 3, Fig. 5. Fig. 6 and Fig. 12.

-          It is not clear how the wind power turbine is modelled. Please clarify the linkage between the wind power turbine model and the formulations of the proposed controllers

-          How the controller parameters in Eq. 12 and Eq. 13 could be selected in the practice?

-          The case study is not realistic. The PV size is 0.5 MW compared to the thermal power generating unit of 2000MW? Please justify the adopted sizes.

-          Please re-write this sentence "The first considered load disturbance is 10% occurred inside the PV plant".

-          Please better describe the source of contingency issues under consideration?

-          Please reproduce Fig. 7 and Fig. 8. The figures are crowded. The critical points are not clear.

-          What is the Y axis of Fig.9. Please provide the legend of the figure.

Author Response

Reviewer 1:- Comments and Suggestions for Authors

This paper proposes a heuristic-based optimization approach to develop load frequency controller to manage frequency constraints during contingency issues in interconnected power systems. The performance of the proposed approach has been compared with other approaches existing in the literature. The presentation of the paper is not adequate. In particular, detailed clarifications and descriptions   of the adopted models, controllers and parameters are needed.

This reviewer has the additional following comments that have to be adequately addressed/discussed:

-          The novelty of this work has to be better illustrated in the Introduction. The description of the proposed approach is also missing in the Introduction.

Regarding to the novelty of the work, it is included at the end of introduction section as follows:

The work contributions can be listed as follows:

  • A new skill optimization algorithm (SOA)-based methodology is proposed to design FOPID-LFC installed with interconnected system with RESs.
  • Two power systems are investigated, PV/Thermal and Thermal/wind turbine/Thermal/PV, at different load disturbances.
  • Excessive comparison to CBOA, JSOA, BO, TDO, and AOS is conducted.
  • Statistical tests of Friedman ANOVA table, Wilcoxon rank test, Friedman rank test, and Kruskal Wallis test are implemented.
  • The competence and reliability of the proposed SOA are confirmed via the fetched results.

Regarding to the proposed approach, section 4 is added.

-          Please add the outline of the paper at the end of the Introduction Section.

This comment is considered in the revised manuscript by adding the following paragraph:

“The paper is organized as follows: section 2 presents the mathematical model of the interconnected system. Section 3 explains the main principle of fractional-order PID controller (FOPID). The proposed skill optimization algorithm is presented in section 4 while section 5 introduces formulation the proposed optimization problem. The numerical analysis is given in section 6 while section 7 handles the conclusions.”

-          The variables in Eq.3 and Eq. 4 are not all defined.

The missing variables are defined as follows:

 is the PV panel output voltage,   represents the simulation time,  is the Boltzmann constant

-          Please provide further details about the adopted parameters in the controller presented in Fig. 3, Fig. 5. Fig. 6 and Fig. 12.

Fig. 3 is just the block diagram of the two interconnected PV/Thermal system without any controllers, however the following paragraphs are added:

Fig. 5:  “The adapted parameters of FOPID controller are kp, ki, kd, λd, and μ, they are identified via the proposed SOA such that the error signal (e(t)) is minimized.”

Fig. 6: “The adapted parameters of the FOPID controller installed with PV plant are kp1, ki1, kd1, λd1, and μ1 while those of the controller of thermal plant are kp2, ki2, kd2, λd2, and μ2.”

Fig. 12: “In such system, four FOPID controllers are installed with the four considered plants, the controller with the first thermal plant has adapted parameters of kp1, ki1, kd1, λd1, and μ1 while the second with wind energy plant (Area 2) has kp2, ki2, kd2, λd2, and μ2. The third controller is installed with the second thermal plant (Area 3), it has adapted parameters of kp3, ki3, kd3, λd3, and μ3. The last controller is installed with the PV plant (Area 4) with parameters of kp4, ki4, kd4, λd4, and μ4. These parameters are identified via the proposed SOA.”

-          It is not clear how the wind power turbine is modelled. Please clarify the linkage between the wind power turbine model and the formulations of the proposed controllers. 

The modelling of WT is clarified by adding the following paragraph:

The transfer function of the wind plant can be written as follows [47]:

                                                                   (9)

where , , and  denote the wind plant gains while  , , and  are the wind plant time constants.

Regarding to the linkage, it is modeled as a block with considering the generated power as output, this is shown in Fig. 13, the author didn’t use any converter circuit for connection.

-          How the controller parameters in Eq. 12 and Eq. 13 could be selected in the practice?

Dear sir, in practice FOPID is a circuit that contains many resistance, capacitance, and op-amps, the parameters given in Eq. 12 and 13 is determined via the optimizer, based on the obtained parameters, the practical parameters can be selected. The following figure shows the practical circuit of op-amp.

-          The case study is not realistic. The PV size is 0.5 MW compared to the thermal power generating unit of 2000MW? Please justify the adopted sizes.

We are sorry sir, there was error in writing, the size of PV is “500 MW” not 500 kW, it is corrected in the revised manuscript.

-          Please re-write this sentence "The first considered load disturbance is 10% occurred inside the PV plant".

It has been rewritten as follows:

“The first considered load disturbance is 10% on the PV plant”

-          Please better describe the source of contingency issues under consideration?

This is considered by adding the following paragraph:

“The source of contingency is sudden change in the load in one area, this leads to high oscillations in the power system frequency and exchange power, this oscillation must be damped via LFC.”

-          Please reproduce Fig. 7 and Fig. 8. The figures are crowded. The critical points are not clear.

Fig. 8 and Fig. 9 are redrawn and become clearer in the revised manuscript

-          What is the Y axis of Fig.9. Please provide the legend of the figure.

The Y-axis title is added but the legend is not available as the figure is generated from the command of ANOVA Wallis, it is not valid to add legend of this graph:

However, the following paragraph is added to explain the figure

“Regarding to the boxplots given in Fig. 10, the penguins obtained via the proposed SOA, TDO, and CBOA have the smallest flippers, while the others have larger flippers. This confirms that, CBOA, TDO, and the proposed SOA reject the null hypothesis. “

Reviewer 2 Report

This manuscript needed more improvement.

Author Response

Reviewer 2:

This manuscript proposes a new metaheuristic-based approach of skill optimization algorithm (SOA) to design fractional-order proportional integral derivative LFC controller installed with multi-interconnected system with multi sources. The integral time absolute error (ITAE) is taken as the frequency and exchange power violations. Two power systems are investigated, the first one has two connected plants of photovoltaic (PV) and thermal units, the second system contains four plants of PV, wind turbines, and two thermal plants with governor dead-band and generation rate constraints, different load disturbances are analysed in both considered systems. Regarding this study, I have few technical queries that has been mentioned below.

(1) The detail of the proposed algorithm should be shown.

This comment is considered by adding section 4.

(2) Why this integral time absolute error (ITAE) objective function is taken into study, explain it in detail.

This comment is clarified in the revised manuscript by adding the following paragraph:

“ITAE integrates the time multiplied by absolute error over a specified time, the ITAE tuning crops systems that settle abundant more rapidly than the other tuning methods that used integral absolute error (IAE) and integral square error (ISE).”

(3) At one case, 10% Load is taken and in different case, 1% Load is taken. The query is 10% load perturbation is too much. LFC problem is a small signal analysis, in this condition, how 10% load perturbation can be applied to this system.

Dear sir, many reported approaches applied 10% disturbance for two-interconnected system, the authors followed these references. However, the most important issue is that vanishing the oscillations in frequency and exchange power deviations after the disturbance and this achieved by the designed LFC.

(4) By seeing Fig. 13, settling time is 161.438 sec for the SOA Method for delf1. The query is if system settle after 161.438 sec, what is the use of this study. Our requirement is to settle the system as early as possible after the load perturbation. Is it feasible time from the LFC perspective.

Dear sir, if you see the settling time of the other considered approaches you will find their settling times are very large. However, the authors added the future recommendation in the conclusion section as follows:

“Accelerating the response of LFC-FOPID controllers in multi-interconnected area via hybrid optimization approach is recommended in the future work.”

(5) In Fig. 14, the Pattern of variable disturbance is shown. The query is why the margin of 200 second is taken for the study. Such a large slot cannot justify the study in practical point of view.

Fig. 15 is assumed pattern of wind speed, the pattern has three sections, the authors take the time margin of each section as 200 sec. like the first case to guarantee zero oscillation in time responses of ΔFi and ΔPtie-i. This is clarified in the obtained responses of Fig. 16. 

(6) Give more details of the fractional-order PID controller. How it is implemented in this system.

The following paragraphs are added in the revised manuscript:

“FOPID is different from the conventional PID as the order of its integral and derivative is not integer. This gives the controller more freedom in the controller tuning, this action results in better dynamic performance of FOPID compared to the conventional one.”

“The FOPID controller is simulated in Simulink via FOMCON Toolbox, it cares about the fractional-order calculus to model, design, and control the system. There are block set provided by this toolbox from which PIλDμ (FOPID) controller.”

(7) Fig. 7 showed the fitness value versus the iteration number for PV/Thermal system at ΔPL1=10%. By seeing this curve, how we can say the proposed method is better than the other method. Almost every curve has the same convergence rate.

Zoomed figure is added inside Fig. 8 that shows the performance of each optimizer during iterative process.

(8) The minimum and maximum limits of the scaling variable is set in range [0.1-2]. By seeing Table 1 and Table 4, no one gain cross 1, why so, is it optical gain.

Dear sir, the parameters tabulated in Table 1 and Table 4 are getting from the optimizers, the authors haven’t any control on them as we followed the upper and lower limits given in previous reported work. However, many gains are identified as unity, there is no rules about optical gain or not. As, we said these values are from the optimizers not from us.

Reviewer 3 Report

1.       Similarity index from below study is very high (%7) it must be reduced;

A.      Fathy and A. G. Alharbi, "Recent Approach Based Movable Damped Wave Algorithm for Designing Fractional-Order PID Load Frequency Control Installed in Multi-Interconnected Plants With Renewable Energy," in IEEE Access, vol. 9, pp. 71072-71089, 2021, doi: 10.1109/ACCESS.2021.3078825.

Also, what is the difference from this study must be shown clearly

2.       The keywords must be increased and revised

3.       There is required a graphical abstract figure to define the manuscript

4.       There is a required at last paragraph for the introduction

5.       What is the GMP in PV plan model it must be explained detailed.

6.       TheHe typo and grammar errors must be checked. For example, the instead of Dc to Dc must be DC to DC.

7.       More papers must be read for more details on PV and wind models. For example;

…….. (2001). Evaluating MPPT converter topologies using a MATLAB PV model. Journal of Electrical & Electronics Engineering, Australia, 21(1), 49-55.

……. (2016). Physical structure, electrical design, mathematical modeling and simulation of solar cells and modules. Turkish Journal of Electromechanics and Energy, 1(1).

……. (2012). A novel filter compensation scheme for single phase-self-excited induction generator micro wind generation system. Scientific Research and Essays, 7(34), 3058-3072.

8.       Figure 3 must be revised. Some parts are not defined. For example, what is Eq. 3 and 4?

9.       Figure 4 is very simple and does not give any information. What is the eras here?

10.   Some of the equation parameters are not defined they must be shown in a table such as a nomenclature. or must be shown with blocks such as PID control.

11.   The SAO algorithm and flowchart must be given.

12.   Figure 6 must be revised and the texts must be readable on %100 screen.  The two parts can be shown differently.

13.   Are there some references for tables?

14.   Figure 8 must be enlarged to see the small plots in it.

15.   Figure 9 is not understandable. It must be explained. What is the axis?

16.    What about the stability analyses of the system? Because you are focused there on graphics.

17.   Figure 12 is not readable. It must be replotted to be read easily. Some parts can be vertical.

18.   All the tables and figures are not commended. Also can be as a results table for the results.

19.   There is no experimental part. Also, the simulation part must be more clear.

20.   The references must be checked for journal format. Also, some current references proposed to read to revise the paper;

………… (2020). A hybrid PV-battery/supercapacitor system and a basic active power control proposal in MATLAB/simulink. Electronics, 9(1), 129.

.......... (2021). Parallel-Connected Buck–Boost Converter with FLC for Hybrid Energy System. Electric Power Components and Systems, 48(19-20), 2117-2129.

Author Response

Reviewer 3: Comments and Suggestions for Authors

  1. Similarity index from below study is very high (%7) it must be reduced;
  2. Fathy and A. G. Alharbi, "Recent Approach Based Movable Damped Wave Algorithm for Designing Fractional-Order PID Load Frequency Control Installed in Multi-Interconnected Plants With Renewable Energy," in IEEE Access, vol. 9, pp. 71072-71089, 2021, doi: 10.1109/ACCESS.2021.3078825.

This comment is considered, the following is a copy of Turnitin, it is clear the mentioned reference comes in the fourth order (1%)

Also, what is the difference from this study must be shown clearly

It is clear that, the algorithm here is (SOA) which is completely different from (DMVA)

  1. The keywords must be increased and revised

The keywords are modified to be:

“LFC; PV plant; wind energy; multi-interconnected system; renewable energy; skill optimization algorithm.”

  1. There is required a graphical abstract figure to define the manuscript

The selected graphical abstract figure is Fig. 7 with title “Fig. 7 Simulink model of PV/Thermal system”

  1. There is a required at last paragraph for the introduction

It is considered in the revised manuscript as follows:

“The paper is organized as follows: section 2 presents the mathematical model of the interconnected system. Section 3 explains the main principle of fractional-order PID controller (FOPID). The proposed skill optimization algorithm is presented in section 4 while section 5 introduces formulation the proposed optimization problem. The numerical analysis is given in section 6 while section 7 handles the conclusions.”

  1. What is the GMP in PV plan model it must be explained detailed.

It is already defined in the manuscript as follows:

“The characteristic of the PV panel is nonlinear, the PV panel power-voltage (P-V) curve has unique global maximum power (GMP) as shown in Fig. 1, it is essential to monitor this point to enhance the PV panel performance and maximize its efficiency.”

  1. The typo and grammar errors must be checked. For example, the instead of Dc to Dc must be DC to DC.

It is corrected in the revised manuscript in addition to all other typos and grammar errors

  1. More papers must be read for more details on PV and wind models. For example;

…….. (2001). Evaluating MPPT converter topologies using a MATLAB PV model. Journal of Electrical & Electronics Engineering, Australia, 21(1), 49-55.

……. (2016). Physical structure, electrical design, mathematical modeling and simulation of solar cells and modules. Turkish Journal of Electromechanics and Energy, 1(1).

……. (2012). A novel filter compensation scheme for single phase-self-excited induction generator micro wind generation system. Scientific Research and Essays, 7(34), 3058-3072.

The mentioned references are cited in the revised manuscript with adding the following sentence:

“Many reported works have been conducted to model PV and WT-based generating systems [38-40].”

  1. Figure 3 must be revised. Some parts are not defined. For example, what is Eq. 3 and 4?

Figure 3 is redrawn and all gains are written in the figure blocks

  1. Figure 4 is very simple and does not give any information. What is the eras here?

Figure 4 shows the second considered interconnected plants, it just general figure of the considered plants in the second case, it is clarified as follows:

“The second considered one is four-connected areas comprising PV, WT, and two thermal areas with GDB and GRC, the architecture of such one is shown in Fig. 4.”

Eras here mean plants

  1. Some of the equation parameters are not defined they must be shown in a table such as a nomenclature. or must be shown with blocks such as PID control.

The Nomenclature table is added before the introduction section with defining all parameters in the manuscript

  1. The SAO algorithm and flowchart must be given.

This comment is considered by adding section 4

  1. Figure 6 must be revised and the texts must be readable on %100 screen.  The two parts can be shown differently.

Figure 7 is redrawn and all blocks are clear now.

  1. Are there some references for tables?

There are no references for the tables, all of them are original.

  1. Figure 8 must be enlarged to see the small plots in it.

This comment is considered in the revised manuscript, the figure is enlarged.

  1. Figure 9 is not understandable. It must be explained. What is the axis?

This comment is considered, the y-axis title is added and the following paragraph is added:

“Regarding to the boxplots given in Fig. 10, the penguins obtained via the proposed SOA, TDO, and CBOA have the smallest flippers, while the others have larger flippers. This confirms that, CBOA, TDO, and the proposed SOA reject the null hypothesis. “

  1. What about the stability analyses of the system? Because you are focused there on graphics.

Our focus not only on the graphs but also the tables and tabulated fitness value, performance specifications given in table 6. Once the proposed controller achieved the target (ΔFi=0, ΔPtie-i=0), this means system is stable.

  1. Figure 12 is not readable. It must be replotted to be read easily. Some parts can be vertical.

The authors did their best to enlarge this figure and clarify all blocks as possible.

  1. All the tables and figures are not commended. Also, can be as a results table for the results.

The authors did their best to redraw the figures based on your recommendations. Also, the tables are organized and include valuable findings.

  1. There is no experimental part. Also, the simulation part must be more clear.

The authors are sorry for this comment, how can experimental setup is conducted on large plants with different sources, we think this need some permission from the electricity sector and this is difficult. We believe that the reviewer is very prejudiced on the work and the analysis included in the work. The simulation part is mentioned in details, also excessive cases are conducted, what must have to do for you satisfaction?

  1. The references must be checked for journal format. Also, some current references proposed to read to revise the paper;

………… (2020). A hybrid PV-battery/supercapacitor system and a basic active power control proposal in MATLAB/simulink. Electronics, 9(1), 129.

.......... (2021). Parallel-Connected Buck–Boost Converter with FLC for Hybrid Energy System. Electric Power Components and Systems, 48(19-20), 2117-2129.

The following paragraph is added in the revised manuscript:

“A power control approach of hybrid renewable energy system has been introduced in [36]. A parallel buck-boost converter controlled via Fuzzy logic control has been constructed and installed with hybrid renewable energy-based system [37].”

Round 2

Reviewer 1 Report

Most of the comments raised in the first round of review have been addressed.  However, the conclusions should be read as standalone document so please avoid using abbreviations. Also, please clearly provide briefly the objectives and the descriptions of the proposed approach (as requested in the first round of review).

Author Response

Reviewer-1: Comments and Suggestions for Authors

Most of the comments raised in the first round of review have been addressed. 

Thankyou sir

However, the conclusions should be read as standalone document so please avoid using abbreviations.

Tis comment is considered, all abbreviations in conclusion section are removed

Also, please clearly provide briefly the objectives and the descriptions of the proposed approach (as requested in the first round of review).

The following paragraph is added at the end of Introduction section

“The aim of this work is to design FOIPD controller based LFC via recent approach of skill optimization algorithm (SOA). The algorithm is responsible for identifying the unknown parameters of the considered controller such that the integral time absolute error of the frequency and exchange power deviations is minimized.”

Reviewer 2 Report

The revised Manuscript is improved a lot in terms of technically and providing more information.

Author Response

Reviewer-2: Comments and Suggestions for Authors

The revised Manuscript is improved a lot in terms of technically and providing more information.

Thankyou sir for your decision

Reviewer 3 Report

The authors complete some of the revisions, but did not have a response to the review file. So some of the comments are not answered satisfied as below;  

1.       There is required a graphical abstract figure to define the manuscript

2.       Figure 4 is very simple and does not give any information. What is the areas here?

3.       Figure 7 must be enlarged.

4.       What about the stability analyses of the system? Bode Nyquist plots?

5.       Figure 13 is not readable. It must be replotted to be read easily. Some parts can be vertical.

6.       All the tables and figures are not commended. Also can be as a results table for the results.

7.       There is no experimental part. Also, the simulation part must be more clear.

Author Response

Reviewer-3: Comments and Suggestions for Authors

The authors complete some of the revisions, but did not have a response to the review file. So some of the comments are not answered satisfied as below;  

  1. There is required a graphical abstract figure to define the manuscript

Graphical abstract is added in the revised manuscript

  1. Figure 4 is very simple and does not give any information. What is the areas here?

Figure 4 shows the second considered interconnected plants, it just general figure of the considered plants in the second case. However, it is changed in the revised manuscript.

  1. Figure 7 must be enlarged.

It is enhanced as much as possible

  1. What about the stability analyses of the system? Bode Nyquist plots?

This is will be considered in the future works. This is clarified by adding the following sentence in conclusion section:

“Moreover, investigating the stability of the system will be considered in the next paper.”

  1. Figure 13 is not readable. It must be replotted to be read easily. Some parts can be vertical.

It is enhanced as much as possible

  1. All the tables and figures are not commended. Also, can be as a results table for the results.

The authors think the figures in the revised manuscript are improved by replotting Figs. 4, 7, and 13

  1. There is no experimental part. Also, the simulation part must be clearer.

Regarding to experimental data, it is required large plants with different sources, the authors think this need some permission from the electricity sector and this is difficult. The simulation part is mentioned in details, also excessive cases are conducted, what must have to do for you satisfaction. However, the authors will think to do this in the future.
